# USP8 inhibition reshapes an inflamed tumor microenvironment that potentiates the immunotherapy

Wenjun Xiong[1,2,11], Xueliang Gao[3,11], Tiantian Zhang[2], Baishan Jiang [2,4], Ming-Ming Hu[2,5], Xia Bu[6], Yang Gao[7,8], Lin-Zhou Zhang[2,9], Bo-Lin Xiao[2,9], Chuan He[1,2], Yishuang Sun[1,2], Haiou Li[2,10], Jie Shi[1,2], Xiangling Xiao[1,2], Bolin Xiang[1,2], Conghua Xie [1], Gang Chen [2,9], Haojian Zhang[2], Wenyi Wei [8], Gordon J. Freeman [6], Hong-Bing Shu [2,5], Haizhen Wang [3✉] & Jinfang Zhang [1,2✉]

Anti-PD-1/PD-L1 immunotherapy has achieved impressive therapeutic outcomes in patients with multiple cancer types. However, the underlined molecular mechanism(s) for moderate response rate (15–25%) or resistance to PD-1/PD-L1 blockade remains not completely understood. Here, we report that inhibiting the deubiquitinase, USP8, significantly enhances the efficacy of anti-PD-1/PD-L1 immunotherapy through reshaping an inflamed tumor microenvironment (TME). Mechanistically, USP8 inhibition increases PD-L1 protein abundance through elevating the TRAF6-mediated K63-linked ubiquitination of PD-L1 to antagonize K48-linked ubiquitination and degradation of PD-L1. In addition, USP8 inhibition also triggers innate immune response and MHC-I expression largely through activating the NF-κB signaling. Based on these mechanisms, USP8 inhibitor combination with PD-1/PD-L1 blockade significantly activates the infiltrated CD8+ T cells to suppress tumor growth and improves the survival benefit in several murine tumor models. Thus, our study reveals a potential combined therapeutic strategy to utilize a USP8 inhibitor and PD-1/PD-L1 blockade for enhancing anti-tumor efficacy.

[1] Department of Radiation and Medical Oncology, Hubei Key Laboratory of Tumor Biological Behaviors, Hubei Cancer Clinical Study Center, Zhongnan Hospital of Wuhan University, 430071 Wuhan, China. [2] Frontier Science Center for Immunology and Metabolism, Medical Research Institute, School of Medicine, Wuhan University, 430071 Wuhan, China. [3] Department of Cell and Molecular Pharmacology & Experimental Therapeutics, Hollings Cancer Center, Medical University of South Carolina, Charleston, SC 29425, USA. [4] Center for Protein Degradation, Dana-Farber Cancer Institute, Harvard Medical School, Boston, MA 02115, USA. [5] Department of Infectious Diseases, Zhongnan Hospital of Wuhan University, 430071 Wuhan, China. [6] Department of Medical Oncology, Dana-Farber Cancer Institute, Harvard Medical School, Boston, MA 02115, USA. [7] Department of Urology, The First Affiliated Hospital of Xi'an Jiaotong University, 710061 Xi'an, China. [8] Department of Pathology, Beth Israel Deaconess Medical Center, Harvard Medical School, Boston, MA 02115, USA. [9] The State Key Laboratory Breeding Base of Basic Science of Stomatology (Hubei-MOST) & Key Laboratory of Oral Biomedicine Ministry of Education and Department of Oral and Maxillofacial Surgery, School and Hospital of Stomatology, Wuhan University, 430071 Wuhan, China. [10] Department of Dermatology, Zhongnan Hospital of Wuhan University, 430071 Wuhan, China. [11]These authors contributed equally: Wenjun Xiong, Xueliang Gao. ✉email: wangha@musc.edu; jinfang_zhang@whu.edu.cn

Cancer immunotherapies, especially targeting the programmed death 1/programmed death-ligand 1 (PD-1/PD-L1) pathway, have achieved impressive therapeutic outcomes in patients with multiple cancer types[1,2]. However, the underlined molecular mechanism(s) for moderate response rate (15–25%) or resistance to PD-1/PD-L1 blockade remains not completely understood[3,4]. Increasing evidence reveals that expression levels of PD-L1, intact antigen presentation, high cytotoxic T lymphocytes (CTLs) infiltration, or interferon (IFN) signaling activation in tumor cells or tumor microenvironment (TME) might be potential hallmarks for better response to PD-1/PD-L1 blockade[5,6]. Thus, thoroughly understanding regulatory mechanisms for PD-L1 and other hallmarks might help overcome the bottleneck of anti-PD-1/PD-L1 immunotherapies through designing combined therapeutic strategies. To this end, recent studies from our and other groups have shown that regulation of immunotherapy responsive hallmarks, including PD-L1, IFN signaling, or major histocompatibility complex class I (MHC-I)-mediated antigen presentation can affect the efficacy of PD-1/PD-L1 blockade in preclinical mouse models and some proposed combinational therapeutic strategies are being tested in clinical trials[7–10].

Ubiquitination is an important type of post-translational modification (PTM) and plays a critical role in regulating various cellular processes through governing protein stability, trafficking, localization, and interaction[11,12]. One ubiquitin molecule has seven lysine (K) residues (K6, K11, K27, K29, K33, K48, and K63), which can be assembled into eight different ubiquitin chain linkages by covalently conjugating the C-terminal glycine of a second ubiquitin molecule with one of the seven lysine residues or the amino-terminal methionine (Met1) on the first ubiquitin moiety. Different ubiquitin chain linkages execute distinct cellular functions[13,14]. It is well-characterized that the K48- or K11-linked ubiquitin chain serves as a destruction signal to trigger 26S proteasome-mediated proteolysis[15], whereas the K63-linked ubiquitination plays a non-degradative signal in NF-κB activation and immune response[16]. Recent studies demonstrated that several E3 ligases destabilize PD-L1 mainly through 26S proteasome- or lysosome-mediated degradation[17–20]. However, whether PD-L1 can be modified by other non-degradative ubiquitin chains to control its physiological functions remains incompletely understood.

In contrast to ubiquitin E3 ligases that conjugate ubiquitin chains on their target proteins, deubiquitinating proteases (DUBs) can cleave and remove ubiquitin chains from their substrate proteins[21,22]. In mammals, more than 100 DUBs have been discovered and the ubiquitin-specific proteases (USP) are the largest subfamily of DUBs with ~54 members[21]. USP8 (also named UBPY) is one member of the USP subfamily and plays an important role in controlling endocytosis and protein trafficking largely through its deubiquitinating activity in regulating the endosomal sorting complexes required for transport (ESCRT)[23,24]. Previous studies also showed that USP8 is frequently overexpressed in human cancers and cancer patients with high USP8 expression have shown worse overall survival[25–27]. Moreover, somatic gain-of-function USP8 mutations with hyper-deubiquitinase activity have been identified in ~50% Cushing's disease, which is caused by adrenocorticotropic hormone (ACTH)-secreting pituitary adenoma[28,29]. Hence, inhibition of USP8 might be a promising therapeutic strategy to USP8-mutated corticotrophin adenoma. Additionally, USP8 was identified as an immunomodulatory DUB and T-cell-specific Usp8-deficient mice developed inflammatory bowel disease largely through disrupting regulatory T-cell functions and recruiting abundant CD8+ γδT cells in colons[30]. Together, these studies demonstrate that USP8 might play a critical role in promoting tumorigenesis and suppressing CD8+ T-cell function, which highlights USP8 could be a potential therapeutic target

in human cancers. However, whether USP8 is involved in regulating cancer immunotherapy has not been reported.

In this study, we uncover a molecular mechanism that USP8 regulates PD-L1 K63-linked ubiquitination and immune response signaling pathways to control anti-tumor immunity. Inhibiting USP8 by depletion or pharmacological inhibitor increases the PD-L1 expression level largely through elevating the TRAF6-mediated K63-linked ubiquitination to antagonize K48-linked ubiquitination and degradation of PD-L1. Moreover, USP8 inhibition also triggers innate immune response including IFN type I signaling activation as well as MHC-1 expression through activating TRAF6-NF-κB signaling, which might counterbalance the adverse effect of PD-L1 expression and set up an inflamed TME where anti-PD-1/PD-L1 immunotherapy can be more effective.

## Results

**USP8 inhibition elevates PD-L1 protein abundance in cancer cells.** Accumulating evidence has shown that DUBs play critical roles in the development of human diseases including cancer. Small-molecule inhibitors targeting the enzymatic activity of DUBs have been developed and are moving forward into preclinical studies or clinical trials[21]. To identify whether DUB inhibitors are involved in regulating the expression of the immune checkpoint protein PD-L1 and the efficacy of PD-1/PD-L1 blockade-based cancer immunotherapy, we screened a panel of DUB inhibitors and discovered that the USP8 inhibitor, DUBs-IN-2, but not other DUB inhibitors we examined, dramatically elevated PD-L1 protein abundance in different cancer cell lines (Fig. 1a and Supplementary Fig. 1a). Moreover, DUBs-IN-2 treatment upregulated the PD-L1 protein level in a dose-dependent manner in multiple cancer cell lines, but did not affect expression levels of other immune checkpoints we examined (Fig. 1b–g and Supplementary Fig. 1b–g). Cell surface PD-L1 on H460 or PC9 cells was also significantly elevated with DUBs-IN-2 treatment (Fig. 1c, d, f, and g). In contrast to PD-L1 upregulation, epidermal growth factor receptor (EGFR) and epidermal growth factor receptor-3 (ErbB3) were downregulated in high-dose DUBs-IN-2 treated PC9 cells (Fig. 1e), which is consistent with previous reports that USP8 stabilizes EGRF and ErbB3 in cells and in vivo[31,32].

To further confirm whether USP8 plays a critical role in negatively regulating PD-L1 protein abundance, we applied genetic methods to deplete endogenous USP8 in cells. Consistent with USP8 inhibition by the DUBs-IN-2 inhibitor, depletion of endogenous USP8 using two independent sgRNAs or shRNAs resulted in a dramatic upregulation of PD-L1 protein levels in various cancer cell lines including CT26 and DLD1, but did not affect the PD-1 expression level in MOLT4 cells (Fig. 1h–j and Supplementary Fig. 1h–j). Cell surface PD-L1 was also significantly upregulated in sgUsp8 cells compared with sgControl cells, whereas there was not significant change on the mRNA level of PD-L1 (Fig. 1i–k). Importantly, DUBs-IN-2 treatment elevated the PD-L1 protein level in sgControl, but not in sgUsp8 cells (Supplementary Fig. 1k), suggesting DUBs-IN-2-mediated upregulation of PD-L1 is largely dependent on the USP8 genetic status. In keeping with the results that USP8 deficiency stabilized PD-L1, ectopic expression of USP8 decreased the PD-L1 protein abundance, but not other immune checkpoints we examined in cells (Supplementary Fig. 1l, m).

As depletion of USP8 dramatically elevated the total and membrane PD-L1 protein abundance, but did not significantly affect the PD-L1 mRNA level (Fig. 1h–k), we speculated that USP8 regulates the PD-L1 protein stability largely at the posttranslational level. There are two major systems to regulate the protein

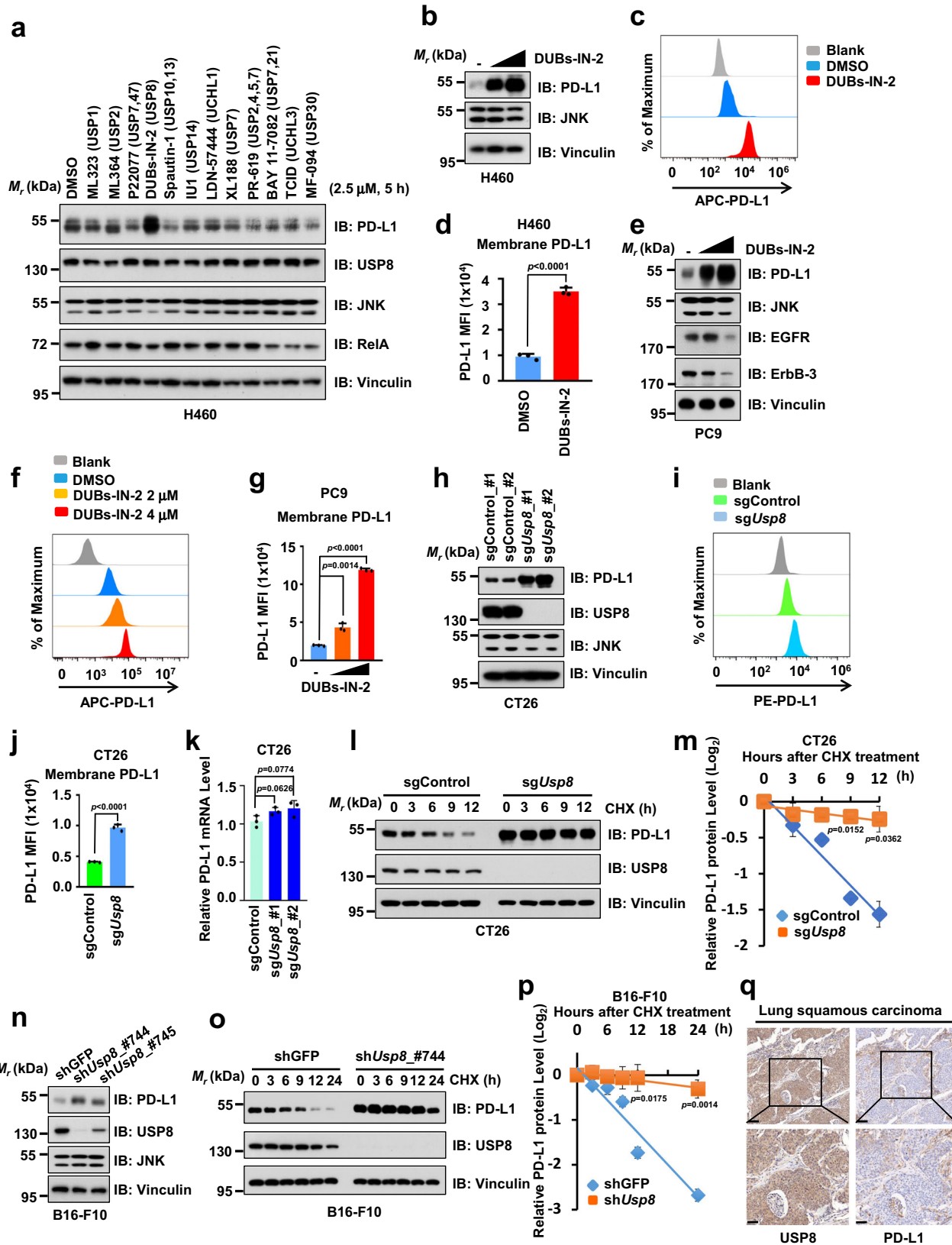

degradation in cells: the proteasome-mediated degradation system and the autophagy-lysosome system. To explore which system plays a major role in regulating the PD-L1 protein stability in our experimental condition, we used the proteasome inhibitor MG132 and the lysosome inhibitor bafilomycin A1 (BafA1) to treat cells and found that the MG132, but not BafA1, alleviated the difference

of PD-L1 expression between sgControl and sg*Usp8* CT26 cells (Supplementary Fig. 1n). Moreover, MG132, but not BafA1 treatment, could efficiently block the ectopic expression of USP8-mediated decrease of PD-L1 in PC9 cells (Supplementary Fig. 1o). Furthermore, MG132 treatment dramatically promoted PD-L1 ubiquitination compared with that of the BafA1 treatment in cells

**Fig. 1 USP8 inhibition elevates PD-L1 protein abundance in cancer cells. a** Immunoblot (IB) analysis of whole-cell lysates (WCL) derived from H460 cells treated with indicated inhibitors or dimethyl sulfoxide (DMSO). Three independent biological repeats were conducted. **b–d** IB analysis of WCL derived from H460 cells treated with DUBs-IN-2 (2 μM and 4 μM) for 6 h (**b**). Cell surface PD-L1 was analyzed after DUBs-IN-2 (2 μM) treatment for 6 h (**c, d**). **e–g** IB analysis of WCL derived from PC9 cells treated with DUBs-IN-2 (2 μM and 4 μM) for 24 h (**e**). Cell surface PD-L1 was analyzed after 24 h for indicated treatment (**f, g**). **h–k** IB analysis of WCL derived from CT26 cells infected with indicated lentiviral sgControl or sg*Usp8* (**h**). Cell surface PD-L1 on indicated CT26 cells was analyzed (**i, j**). PD-L1 mRNAs were analyzed using reverse transcription quantitative PCR (RT-qPCR) (**k**). **l, m** IB analysis of WCL derived from sgControl- or sg*Usp8*-treated CT26 cells treated with 400 μg/ml cycloheximide (CHX) at indicated time points (**l**). PD-L1 band intensity was quantified by ImageJ, which was normalized to vinculin and then to the t = 0 time point (**m**). **n–p** IB analysis of WCL derived from shGFP- or sh*Usp8*-treated B16-F10 cells, which were selected with puromycin (1 μg/ml) for generating stable cell lines; three independent biological repeats were conducted (**n**). IB analysis of WCL derived from B16-F10 cells stably infected with indicated lentiviral shRNAs. Cells were treated with 200 μg/ml CHX at indicated time points (**o**). PD-L1 band intensity was quantified by ImageJ, which was normalized to vinculin and then to the t = 0 time point (**p**). **q** Representative images from IHC staining of PD-L1 and USP8 in human lung squamous carcinoma. Scale bar, upper panels: 100 μm; lower panels: 50 μm. n = 63 biologically independent patient samples. For **d**, **g**, **j**, **k**, **m**, and **p** data were presented as mean ± S.D.; n = 3 biologically independent samples; Two-sided t-test. The relevant raw data and uncropped dots are provided as a Source Data file.

(Supplementary Fig. 1p, q). These results suggest that USP8 mainly utilizes the proteasome system to control the PD-L1 protein stability in cells. Next, we utilized cycloheximide (CHX) to inhibit protein translation and analyzed the protein half-life of PD-L1. Compared to the control cells, the protein half-life of endogenous PD-L1 was dramatically prolonged in sh*Usp8*- or sg*Usp8*-treated cells (Fig. 1l–p). However, the difference in the protein half-life of PD-L1 between WT and *Usp8*-deficient CT26 cells was almost disappeared upon MG132 treatment (Supplementary Fig. 1r, s). Notably, immunohistochemistry (IHC) staining results showed that the USP8 had a negative correlation with PD-L1 in samples of human lung squamous cancer patients (Fig. 1q and Supplementary Fig. 1t), further supporting the notion that USP8 negatively regulates PD-L1 protein stability. Taken together, these results suggest that USP8 inhibition by either the pharmacological inhibitor DUBs-IN-2 or genetic depletion could dramatically elevate the PD-L1 protein abundance largely at the posttranslational level in cancer cells.

**USP8 specifically interacts with PD-L1 to remove its K63-linked poly-ubiquitination.** Previous studies have shown that USP8 not only stabilizes its downstream substrates but also can decrease the substrate largely through its deubiquitinating enzymatic activity, which might be dependent on removing which type of ubiquitin-linked chain from the substrate[31–34]. Our results above have demonstrated that USP8 inhibition dramatically elevates PD-L1 protein levels at a posttranslational stage. In order to test whether USP8 directly interacts with PD-L1 to control its ubiquitination status and stability, we examined the interaction of PD-L1 with a panel of DUBs, most of which are targets of the DUB inhibitors tested in Fig. 1a. We observed that USP8, but not other DUBs we examined, specifically interacted with PD-L1 in cells (Fig. 2a). Moreover, the interaction between PD-L1 and USP8 was observed at endogenous levels in multiple cell lines (Fig. 2b, c and Supplementary Fig. 2a). The glutathione S-transferase (GST) pull-down assay showed that bacterially purified recombinant GST-USP8, but not GST protein, interacted with PD-L1 (Fig. 2d). Furthermore, we explored the critical region(s) within USP8 that binds to PD-L1 in cells. To this end, we truncated USP8 into three major regions: N-terminal domain (amino acid (aa) 1–313 containing MIT and Rhodanese domain), middle region (aa314–714), and C-terminal domain (aa715–1118 containing USP domain) (Fig. 2e). Our results showed that both the N-terminal and C-terminal domains, but not the middle region of USP8, interacted with PD-L1 in cells (Fig. 2e, f). Moreover, the N-terminal domain of USP8 had a relatively higher binding affinity to PD-L1 compared with the C-terminal domain of USP8 (Fig. 2f). In addition, our results showed that PD-L1 interacted with USP8 largely through its

cytoplasmic tail (C-tail, aa260–290) as the PD-L1 deleting the C-tail (PD-L1 ΔC-tail) mutant failed to bind with USP8 in cells (Supplementary Fig. 2b, c).

USP8 executes its physiological functions mainly through the deubiquitinating enzyme activity to antagonize a K48- or K63-linked ubiquitination. A recent study of screening DUB activity and specificity showed that USP8 preferred cleavage of K63-linked ubiquitin chain to K48-linked ubiquitin chain[35]. To examine whether USP8 affects the ubiquitination of PD-L1, we performed an in vivo de-ubiquitination assay in cells and found that the wild type USP8 (USP8-WT), but not the enzymatically inactive mutant USP8-C786A[36], dramatically inhibited the ubiquitination of PD-L1, suggesting that USP8-mediated regulation of PD-L1 might depend on the enzymatic activity of USP8 (Fig. 2g). It is well-known that the K48-linked ubiquitin chain is a signal for trigging 26 S proteasome-mediated proteolysis[21,22]. Our results above demonstrated that USP8 negatively regulated PD-L1 protein abundance, indicating that USP8 might not cleave the canonical K48-linked ubiquitination on PD-L1 since this would be expected to positively regulate PD-L1 protein abundance.

To determine which type of ubiquitin chain linkage(s) was assembled on PD-L1, we co-transfected PD-L1 with each of seven ubiquitin K-only constructs that each kept only the one indicated lysine while the remaining six lysine residues were mutated to arginine (Fig. 2h). In addition to the K48-linked ubiquitination, PD-L1 was also heavily modified by the K63-linked ubiquitination (Fig. 2h). Moreover, the endogenous K63-linked ubiquitination on PD-L1 was also detected with immunoblotting using the K63-linked ubiquitin chain-specific antibody in multiple cancer cell lines (Fig. 2I, j and Supplementary Fig. 2d, e). Notably, in vivo de-ubiquitination assays showed that USP8 mainly removed the K63-linked ubiquitin chain on PD-L1, accompanied by the elevated K48-linked ubiquitination of PD-L1 (Fig. 2k and Supplementary Fig. 2f–i). These results indicated that the K63-linked ubiquitination might compete with K48-linked ubiquitination on PD-L1 to govern PD-L1 stability in cells. A previous report showed that CSN5 can stabilize PD-L1 by removing the poly-ubiquitination on PD-L1[37]. Our results also demonstrated that unlike USP8 cleaving the K63-linked ubiquitination on PD-L1, CSN5 mainly removed the K48-linked ubiquitination on PD-L1 (Supplementary Fig. 2j, k). Notably, inhibition of USP8 using genetic depletion or pharmacological inhibitor obviously elevated endogenous K63-linked ubiquitination, accompanying with reduced K48-linked ubiquitination of PD-L1 in CT26 cells (Fig. 2l and Supplementary Fig. 2l). On the other hand, stably ectopic expression of USP8 dramatically reduced the endogenous K63-linked ubiquitination of PD-L1 and increased the K48-linked ubiquitination

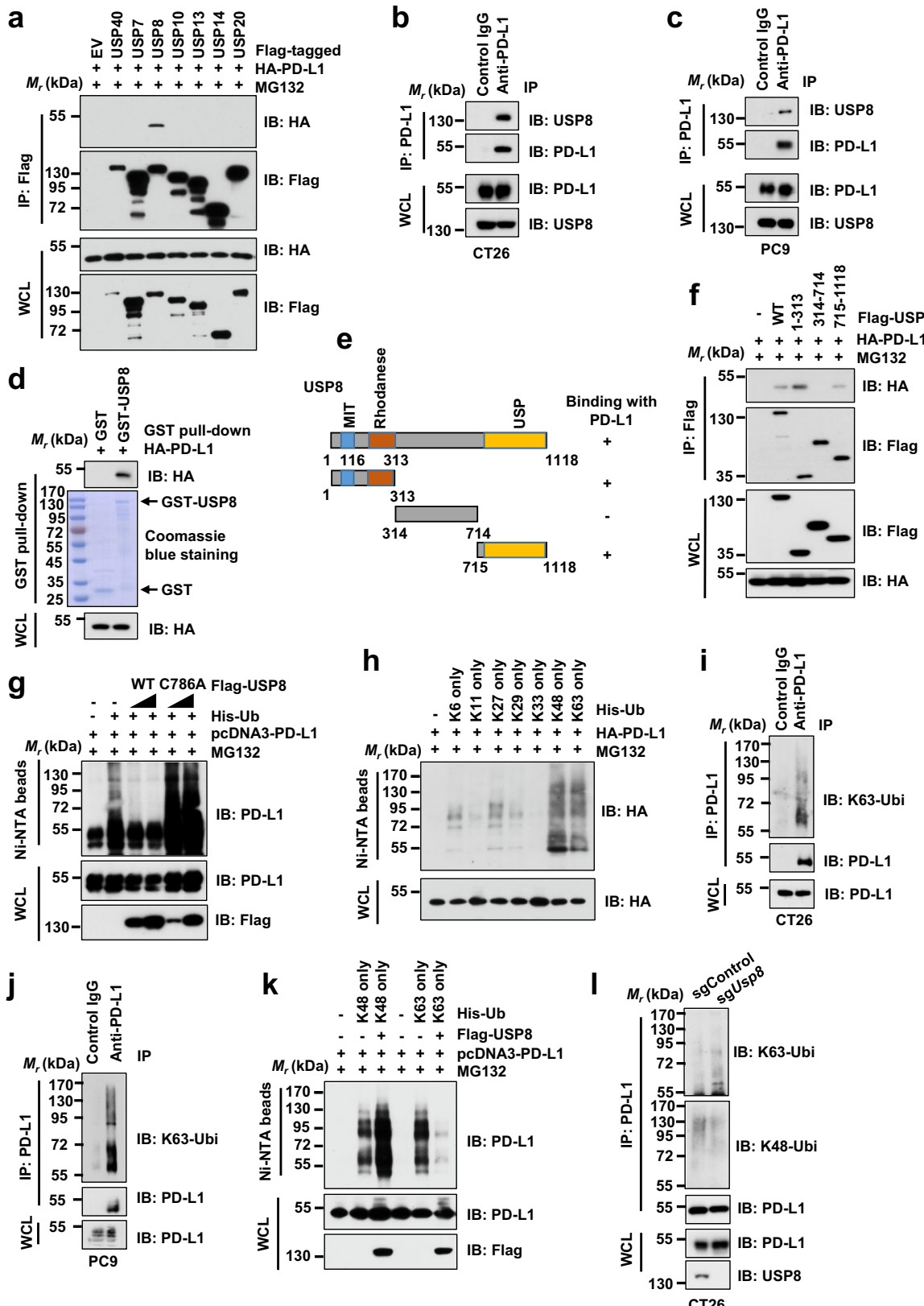

of PD-L1 in CT26 and PC9 cells (Supplementary Fig. 2m, n). These results together suggest that the ubiquitination and de-ubiquitination process of PD-L1 is undergoing a dynamic regulation in cells, with different E3 ligase and DUBs potentially dictating different types of ubiquitination linkage including but not limited to K48 and K63 to impact PD-L1 stability and function. Our results further illustrate that unlike CSN5-mediated stabilization of PD-L1, USP8 negatively regulates PD-L1 protein abundance largely by removing the K63-linked ubiquitination of PD-L1.

**Fig. 2 USP8 specifically interacts with PD-L1 and removes its K63-linked poly-ubiquitination. a** Immunoblot (IB) analysis of whole-cell lysates (WCL) and anti-Flag immunoprecipitates (IPs) derived from 293T cells transfected with indicated constructs. Cells were treated with 10 μM MG132 for 12 h before harvesting. EV: empty vector. **b, c** IB analysis of WCL and anti-PD-L1 IPs derived from CT26 (**b**) and PC9 (**c**) cells. **d** IB analysis of glutathione S-transferase (GST) pull-down precipitates from 293 T-cell lysates with ectopic expression of HA-PD-L1 incubated with bacterially purified recombinant GST or GST-USP8 protein. **e** A schematic illustration of USP8 with different domains including the N-terminal amino acid (aa) 1–313, middle region (aa314–714), and C-terminal USP domain (aa715–1118). MIT: microtubule interacting and transport; USP: ubiquitin-specific peptidase. **f** IB analysis of WCL and anti-Flag IPs derived from 293T cells transfected with indicated constructs. Cells were treated with 10 μM MG132 for 12 h before harvesting. **g, h** IB analysis of WCL and Ni-NTA pull-down products derived from lysates of 293T cells transfected with the indicated constructs. Cells were treated with 10 μM MG132 for 12 h before harvesting. Ub: ubiquitin. **i, j** IB analysis of WCL and anti-PD-L1 IPs derived from lysates of CT26 (**i**) and PC9 (**j**) cells using indicated antibodies. Cells were treated with 20 μM MG132 for 6 h before harvesting. **k** IB analysis of WCL and Ni-NTA pull-down products derived from lysates of 293T cells transfected with the indicated constructs. Cells were treated with 10 μM MG132 for 12 h before harvesting. **l** IB analysis of WCL and anti-PD-L1 IPs derived from lysates of sgControl or sg*Usp8* CT26 cells using indicated antibodies. Cells were treated with 20 μM MG132 for 6 h before harvesting. For **a–d** and **f–l**, two independent experiments were conducted. The relevant uncropped dots are provided as a Source Data file.

**The E3 ligase TRAF6 positively regulates PD-L1 protein abundance through promoting the K63-linked ubiquitination of PD-L1.** Recent reports have shown that the ubiquitin E3 ligase SPOP or β-TRCP destabilizes PD-L1 via 26 S proteasome-mediated degradation[17,18]. However, the E3 ligase(s) that stabilize PD-L1 and promote K63-linked ubiquitination of PD-L1 remains unknown. To this end, we screened a panel of E3 ligases regulating PD-L1 protein abundance using a luciferase reporter assay and found that TRAF6 dramatically upregulated the PD-L1-luciferase activity compared to other E3 ligases we tested (Supplementary Fig. 3a, b), indicating that the E3 ligase TRAF6 might stabilize PD-L1 in cells. In keeping with this notion, ectopic expression of TRAF6 increased PD-L1 protein levels in a dose-dependent manner (Fig. 3a and Supplementary Fig. 3c, d). To further explore whether the E3 ligase activity of TRAF6 is critical to regulate the PD-L1 protein level, we generated the TRAF6-C70A mutant that lacks the E3 ligase activity[38]. Our results showed that the TRAF6-C70A failed to upregulate the PD-L1 protein level compared to TRAF6-WT, indicating that the E3 ligase activity of TRAF6 is essential for regulating PD-L1 protein expression (Fig. 3a). Furthermore, ectopic expression of TRAF6 significantly extended the protein half-life of PD-L1, suggesting the E3 ligase TRAF6 stabilized PD-L1 at the post-translational level (Fig. 3b, c). In keeping with the notion that TRAF6 positively regulated PD-L1 stability, depletion of endogenous *TRAF6* using sgRNAs markedly decreased the PD-L1 protein level, but did not significantly affect the PD-L1 mRNA level in multiple cancer cell lines (Fig. 3d–g and Supplementary Fig. 3e, f). These results collectively indicate that TRAF6 positively regulates PD-L1 at the post-translational level mainly through the E3 ligase activity of TRAF6.

To evaluate the clinical relevance of our findings, we examined TRAF6 and PD-L1 protein expression levels with IHC staining in tissues from human lung squamous cancer patients. A positive correlation was observed between the expression of TRAF6 and PD-L1 among these tumor tissues (Fig. 3h, i), further supporting the notion that TRAF6 positively regulated PD-L1 protein stability. These results together suggest that the TRAF6-PD-L1 signaling axis might play an important role in regulating cancer immune evasion and tumorigenesis.

The E3 ligase TRAF6 catalyzes the formation of the K63-linked ubiquitin chains on several substrate proteins to regulate various cellular signaling pathways, including innate and adaptive immune response pathways[39,40]. Since our results showed that TRAF6 stabilized PD-L1 largely at the post-translational stage (Fig. 3b–g), we speculated that TRAF6 might directly interact with and stabilize PD-L1 through promoting K63-linked ubiquitination of PD-L1. We examined the interaction of PD-L1 with all TRAF family members and Skp2, another E3 ligase reported to catalyze the formation of K63-linked ubiquitin

chain[41]. Intriguingly, only TRAF3 and TRAF6, but not other TRAF family members, nor Skp2, interacted with PD-L1 in cells (Fig. 3j and Supplementary Fig. 3g). While TRAF6 dramatically upregulated PD-L1 protein levels (Fig. 3a–c), TRAF3 only slightly elevated PD-L1 protein level in cells (Supplementary Fig. 3h, i). These results suggest that TRAF6, but not TRAF3, plays a critical role in positively regulating the PD-L1 level in cells. Furthermore, the GST pull-down assay demonstrated that GST-TRAF6, but not GST protein, interacted with PD-L1 (Fig. 3k). To determine which domain(s) on the TRAF6 protein interacts with PD-L1, we generated several truncation mutants of TRAF6 and found that the central coiled-coil region of TRAF6 plays a major role in mediating TRAF6 interaction with PD-L1 in cells (Fig. 3l, m). To further identify which region of PD-L1 interacting with TRAF6, we examined the PD-L1 protein sequence and mapped an evolutionarily conserved putative TRAF6-binding motif (PxExxZ) in the N-terminal region of PD-L1 (Supplementary Fig. 3j)[40]. We generated the PD-L1 P43F/E45N mutant by changing two key amino acid residues (P to F and E to N) in the TRAF6-binding motif on PD-L1 (Supplementary Fig. 3j). Our results demonstrated that the PD-L1 P43F/E45N, but not the PD-L1 ΔC-tail with deleting the cytoplasmic domain, disrupts the binding with TRAF6, suggesting that PD-L1 interacts with TRAF6 largely through the TRAF6-binding motif in PD-L1 (Supplementary Fig. 3k, l).

In agreement with these findings, ectopic expression of TRAF6 dramatically promoted the K63-linked ubiquitination of PD-L1 in cells (Supplementary Fig. 3m, n). However, the E3 ligase inactive mutant TRAF6-C70A failed to catalyze the K63-linked ubiquitination of PD-L1 (Fig. 3n), which is consistent with our results that TRAF6-C70A did not upregulate PD-L1 protein levels in cells (Fig. 3a). Of note, genetic depletion of *Traf6* decreased endogenous K63-linked ubiquitination, accompanying with increased K48-linked ubiquitination of PD-L1 in CT26 cells (Supplementary Fig. 3o). In contrast, ectopic expression of TRAF6 elevated endogenous K63-linked ubiquitination and reduced K48-linked ubiquitination of PD-L1 in CT26 cells (Supplementary Fig. 3p). As USP8 could remove the K63-linked ubiquitin chain on PD-L1, we next sought to examine whether USP8 can remove the TRAF6-mediated K63-linked ubiquitination on PD-L1. Notably, ectopic expression of USP8 dramatically reduced the TRAF6-promoted K63-linked ubiquitination of PD-L1 in cells (Fig. 3o). Taken together, these results pinpointed the coiled-coil domain of TRAF6 as mediating the interaction of TRAF6 with PD-L1 and facilitating the K63-linked ubiquitination of PD-L1, which was antagonized by USP8.

**USP8 deficiency elevates multiple immune response genes that facilitate the anti-tumor immunity.** It has been reported that besides the PD-L1 expression level, other key factors including

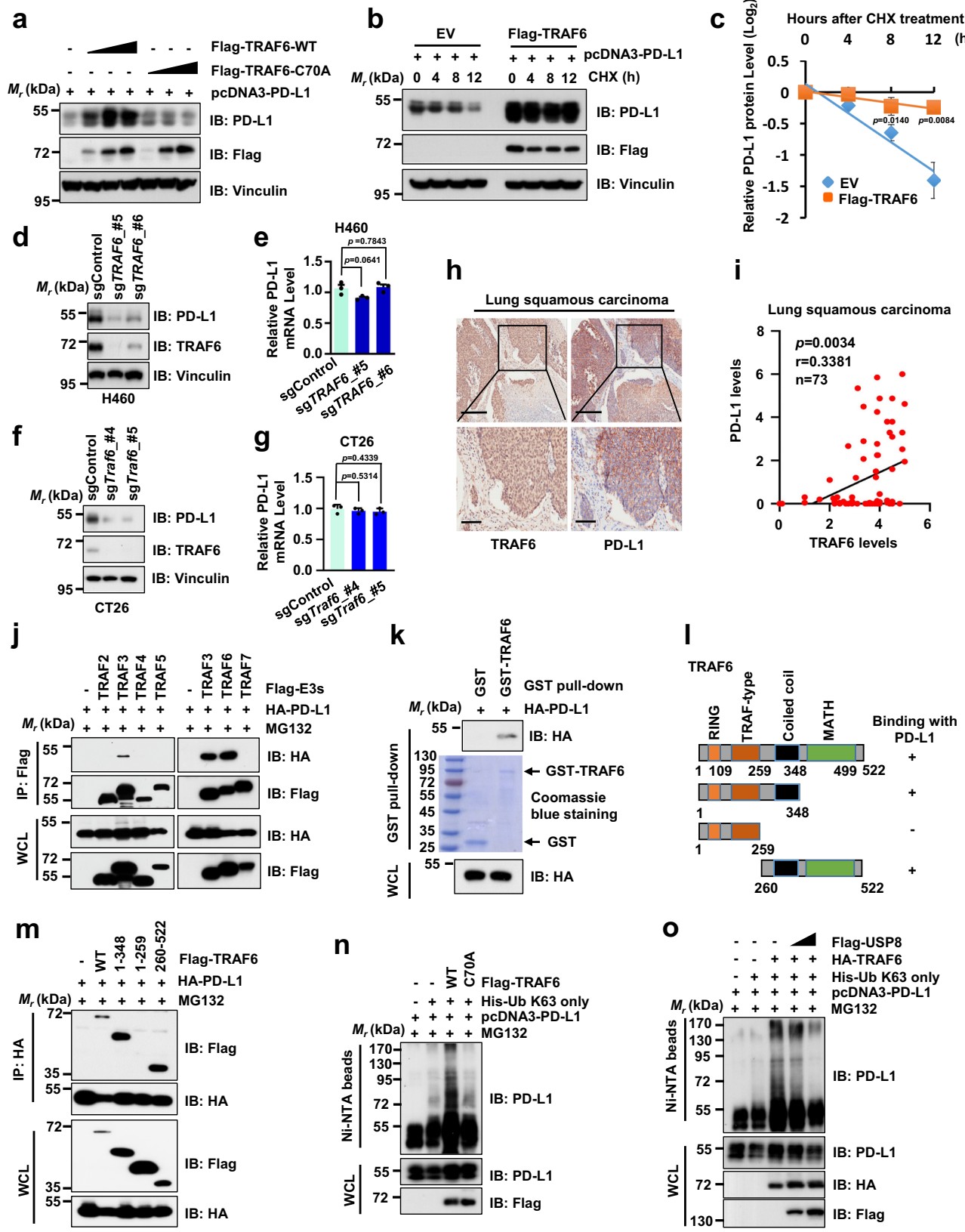

the IFN signaling activation, antigen-presenting and cytotoxic T-cell infiltration also affect the efficacy of immune checkpoint blockade[17,42,43]. To further explore the physiological function of USP8, we performed the transcriptomic analysis to comprehensively understand the signaling pathways are mainly regulated by USP8 in cancer cells. sg*Usp8*- and sgControl-treated CT26 cells were harvested for RNA sequencing (RNA-seq). Surprisingly, gene ontology (GO) and heatmap analysis showed that differentially expressed genes in the top ten enriched biological processes are associated with innate and adaptive immune response in the *Usp8*-deficient cells compared to control cells (Fig. 4a–c). Moreover, most of the upregulated genes in *Usp8*-deficient cells

**Fig. 3 The E3 ligase TRAF6 positively regulates PD-L1 protein abundance through promoting K63-linked ubiquitination of PD-L1. a** Immunoblot (IB) analysis of whole-cell lysates (WCL) derived from 293 T cells co-transfected with indicated constructs. **b, c** IB analysis of WCL derived from 293 T cells co-transfected with indicated constructs. Cells were treated with 200 μg/ml cycloheximide (CHX) as indicated time points (**b**). PD-L1 band intensity was quantified by ImageJ, which was normalized to vinculin and then to the t = 0 time point (**c**). EV: empty vector. **d–g** IB analysis of WCL derived from sgControl or sgTRAF6-treated H460 (**d**) or CT26 cells (**f**). PD-L1 mRNAs were analyzed using the RT-qPCR (**e, g**). **h, i** Representative images from IHC staining of PD-L1 and TRAF6 in human lung squamous carcinoma (**h**). Scale bar, upper panels: 300 μm; lower panels: 100 μm. Quantification of PD-L1 and TRAF6 staining intensities were performed by semi-quantitative scoring (**i**). n = 73, r = 0.3381, p = 0.0034; correlation coefficients were calculated using the Pearson test. Two-sided p-value was given. **j** IB analysis of WCL and anti-Flag IPs from 293T cells co-transfected with indicated Flag-TRAF constructs. **k** IB analysis of glutathione S-transferase (GST) pull-down precipitates from 293T cell lysates with ectopic expression of HA-PD-L1 incubated with bacterially purified recombinant GST or GST-TRAF6 protein. **l** A schematic illustration of TRAF6 protein sequence with different domains or truncated mutants. **m** IB analysis of WCL and anti-HA IPs from 293T cells co-transfected with indicated constructs. **n** IB analysis of WCL and Ni-NTA pull-down products derived from lysates of 293T cells co-transfected with indicated constructs. **o** IB analysis of WCL and Ni-NTA pull-down products derived from lysates of 293T cells co-transfected with indicated constructs. For **j, m, n,** and **o,** cells were treated with 10 μM MG132 for 12 h before harvesting. For **c, e,** and **g** data were presented as mean ± S.D. n = 3 biologically independent samples. Two-sided t-test. For **a, j, k,** and **m–o,** two independent experiments were conducted. The relevant raw data and uncropped dots are provided as a Source Data file.

are genes response to interferon-beta (IFN-β), interferon-gamma (IFN-γ), and virus (Fig. 4a–c). Gene-set enrichment analysis (GSEA) showed that gene expression signatures including the response to interferon-alpha (IFN-α), IFN-β, and IFN-γ were also positively enriched in sgUsp8-treated CT26 cells (Supplementary Fig. 4a–d). Several IFN-sensitive transcription factors (Stat1, Stat2, Irf7), IFN-stimulated genes (Isg15, Oas1, Oas2, Oas3, Ifit1, Ifit2, Ifit3, Bst2), and IFN-inducible T-cell chemo-attractants (Cxcl10, Ccl2, Ccl7, Ccl20) were significantly upregulated in the sgUsp8-treated CT26 cells (Fig. 4a–c). Notably, our results of the reverse transcription quantitative PCR (RT-qPCR) confirmed that a panel of genes response to IFN-α/β/γ and virus were significantly upregulated in the Usp8-deficient CT26 (Fig. 4d) or PC9 cells (Fig. 4e and Supplementary Fig. 4e). These results support a model that Usp8 inhibition in cancer cells might elevate a panel of immune response genes and T-cell chemo-attractants to trigger the anti-tumor immunity.

**USP8 inhibition upregulates the antigen presentation largely through activating the TRAF6-NF-κB signaling.** Further analysis of RNA-seq data showed that a panel of genes involved in the MHC-I-mediated antigen processing and presenting were also positively enriched in the Usp8-deficient CT26 cells (Fig. 5a, b). Moreover, results from our bioinformatic analysis demonstrated that most of the genes in the MHC-I pathways are significantly upregulated in lung or colon adenocarcinoma patients with low USP8 compared with high USP8 expression (Supplementary Fig. 5a, b). We further confirmed that Usp8 depletion significantly elevated gene expression levels of the MHC-I-dependent antigen processing and presentation pathway in CT26 and PC9 cell lines (Fig. 5c–f). Furthermore, the USP8 inhibitor, DUBs-IN-2, also significantly upregulated the expression of MHC-I pathway-related genes in PC9 and H460 cells (Fig. 5g and Supplementary Fig. 5c–e). These results suggested that inhibiting USP8 by genetic depletion or pharmacologic inhibitor increases the antigen processing and presentation, which might support cytotoxic T cells to eliminate cancer cells.

Furthermore, the association between the cytotoxic T lymphocyte (CTL) level and overall survival (OS) for colorectal cancer patients was analyzed under the condition of high or low USP8 expression using Kaplan–Meier curves by the Tumor Immune Dysfunction and Exclusion (TIDE) tool[44]. In the group of cancer patients with low USP8 expression, a higher level of CTL indicated a better survival (Fig. 5h). However, in the group of cancer patients with high USP8 expression, a higher level of CTL showed a worse patient survival, suggesting that a high level of USP8 might lead to T-cell dysfunction (Fig. 5h). Moreover, we also found that ectopic expression of TRAF6 could significantly elevate the expression of MHC-I pathway-related genes (Supplementary Fig. 5f, g). When cancer patients had high

TRAF6 expression, a higher level of CTL indicated a better survival (Supplementary Fig. 5h–j). These results together demonstrated that cancer patients with low USP8 or high TRAF6 had high expression of antigen presentation, which indicated better survival when accompanied by high CTLs infiltration.

Next, we explored the molecular mechanism that USP8 deficiency enhances the immune response and antigen presentation. Previous studies have shown that the K63-linked poly-ubiquitination of TRAF6 is necessary to activate the NF-κB signaling, which is evolutionarily conserved regulators of immune and inflammatory responses[39,45,46]. Our results demonstrated that both TRAF6 and USP8 can interact with PD-L1 in cells (Figs. 2 and 3), suggesting that USP8 might also have the chance to bind with TRAF6 in cells. Indeed, we found that USP8 interacted with TRAF6 largely through the N-terminal domain (aa1–313) of USP8 in cells (Supplementary Fig. 5k). The TRAF6 interacted with USP8 via its central coiled-coil domain (Fig. 5i). Moreover, the interactions among USP8, PD-L1 and TRAF6 might be antagonistic as gradient ectopic expression of any one disrupted the interaction between the other two proteins in cells (Supplementary Fig. 5l–n). Meanwhile, we cannot exclude other mechanisms that affect their interactions. For example, the post-translational modification including ubiquitination on the PD-L1 or TRAF6 might alter their conformation to affect their interactions.

Of note, USP8 inhibited the K63-linked poly-ubiquitination of TRAF6 largely through the deubiquitinase activity of USP8 in cells (Supplementary Fig. 5o). These results suggest that USP8 might inhibit the TRAF6-mediated NF-κB activation largely through removing the K63-linked poly-ubiquitination of TRAF6 in cells. In keeping with these results above, depletion of Usp8 dramatically upregulated the level of phosphorylated p65/RelA (p-p65), indicating activation of TRAF6/NF-κB pathway in CT26 cells (Fig. 5j). Notably, the NF-κB signaling pathway inhibitor, IKKi[47,48], eliminated the upregulation of p-p65 as well as the downstream target genes involving in immune response and antigen presentation in sgUsp8 CT26 cells (Fig. 5j, k and Supplementary Fig. 5p). Furthermore, depletion of endogenous p65 could significantly decrease the MHC-I, but not the PD-L1, at both protein and mRNA levels in sgUsp8 CT26 cells (Fig. 5l–o and Supplementary Fig. 5q, r). Thus, these results demonstrate that USP8 suppresses the immune response and antigen presentation largely through removing the K63-linked poly-ubiquitination of TRAF6, resulting in limiting the activation of NF-κB signaling pathway.

**The combination of USP8 inhibitor with PD-1/PD-L1 block-ade significantly suppresses tumor growth and enhances the survival rate in multiple mouse tumor models.** Our results

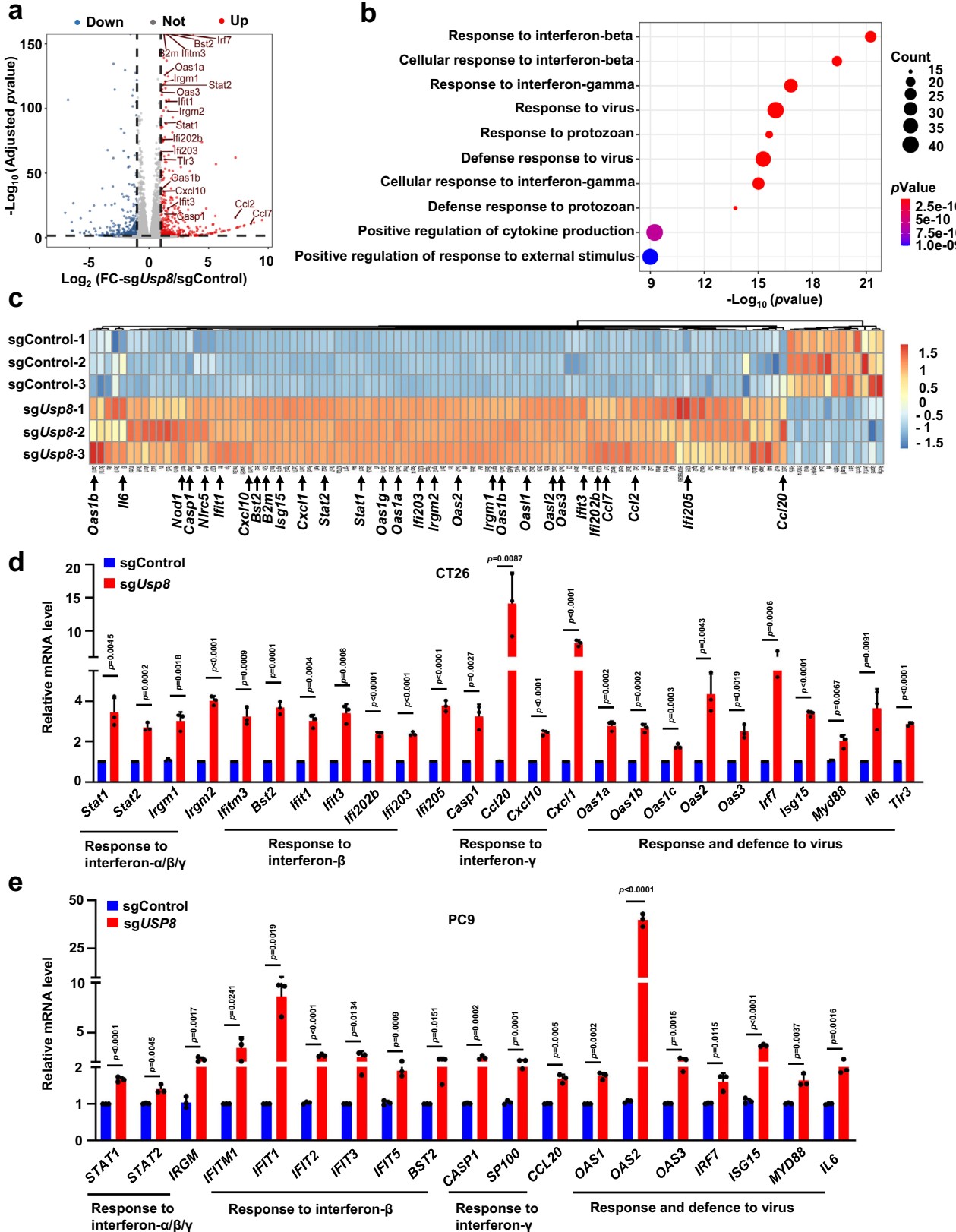

above demonstrated that inhibition of USP8 by genetic depletion or the pharmacological inhibitor, DUBs-IN-2, dramatically upregulates PD-L1 protein levels as well as multiple innate and adaptive immune response signaling pathways that might reshape an inflamed TME to enhance anti-tumor immunity (Figs. 1, 4, and 5). Based on the molecular mechanism study, we hypothesized

that inhibition of USP8 might sensitize tumors to the anti-PD-1/PD-L1 immunotherapy in vivo.

To test this hypothesis, we utilized the syngeneic mouse MC38 tumor model to examine how the combination of USP8 inhibitor, DUBs-IN-2, with anti-PD-L1 antibody affected tumor growth and mice survival. Strikingly, our results showed that combinational

**Fig. 4 *USP8* deficiency elevates multiple immune response genes that facilitate anti-tumor immunity. a** Volcano plot showing differential gene expression for RNA-seq results from sg*Usp8* versus sgControl CT26 cells. Dots in red represent 473 upregulated genes (log$_2$(FC) > 1 and adjusted *p*-value < 0.05) and dots in blue represent 514 downregulated genes (log2(FC) < -1 and adjusted *p*-value < 0.05) in sg*Usp8* versus sgControl CT26 cells. Highlighted genes are involved in innate and adaptive immune response pathways. FC: fold-change. Statistical analysis was performed using Wald-test with Benjamini-Hochberg correction. **b** A dot map showing top 10 terms in Gene Ontology (GO) analysis of differential genes in sg*Usp8* versus sgControl CT26 cells. *n* = 3 biologically independent samples per group. Statistical analysis was performed using modified Fisher's exact tests. **c** Heatmap showing differential expression of genes in the Fig. 4b of top 10 terms in GO analysis. **d**, **e** RT-qPCR analysis of the indicated genes from sgControl and sg*Usp8* CT26 (**d**) or PC9 (**e**) cells. Data were presented as mean ± S.D. *n* = 3 biologically independent samples. Two-sided *t*-test. The relevant raw data and uncropped dots are provided as a Source Data file.

treatment of the USP8 inhibitor plus anti-PD-L1 antibody significantly suppressed tumor growth and improved the overall survival rates of MC38 tumor-bearing immunocompetent C57BL/6 mice compared to either single-agent or control-treated group (Fig. 6a, b and Supplementary Fig. 6a, b). To further confirm this result, we applied another syngeneic mouse tumor model, CT26 tumor-bearing immunocompetent BALB/c mice, to test the combinational effect following the experimental plan (Supplementary Fig. 6c). We also observed that combining the USP8 inhibitor with anti-PD-1 or anti-PD-L1 therapy significantly retarded the CT26 tumor growth and dramatically improved the overall survival compared with either treatment alone (Fig. 6c, d and Supplementary Fig. 6d).

Additionally, we also examined whether the USP8 inhibitor combination with PD-L1 blockade could suppress tumor growth in the autochthonous non-small cell lung cancers (NSCLC) of Kras$^{LSL-G12D/+}$Tp53$^{fl/fl}$ (KP) mice model. Consistent with results from syngeneic mouse colon tumor models (Fig. 6a, c), the USP8 inhibitor combined with anti-PD-L1 antibody significantly suppressed tumor development in KP mice, evidenced by the reduced tumor sizes and areas, compared to either each agent alone or control group (Fig. 6e–g and Supplementary Fig. 6e). Together, our results demonstrate that the combined therapy with the USP8 inhibitor, DUBs-IN-2, and anti-PD-1/PD-L1 has similar efficacy in both lung and colon cancer tumor models, indicating that the mechanism of this study should be suitable for both NSCLC and colon cancers.

Analysis of infiltrated immune cells demonstrated that the USP8 inhibitor combined with anti-PD-L1 treatment could significantly increase the percentage of CD8$^+$ T cells, but not the CD4$^+$ cells in tumor-infiltrating lymphocytes (TILs) (Fig. 6h and Supplementary Fig. 6f). Moreover, there were no significant changes in B cells and dentritic cells after the combined treatment (Supplementary Fig. 6g, h). To further address whether the USP8 inhibitor affects the activation of tumor-infiltrating CD8$^+$ T cells and the profile of exhausted T cells, we also detected the T-cell activation maker, Granzyme B (GzmB), and exhausted T-cell marker, TIM3, on infiltrated CD8$^+$ T cells in syngeneic CT26 mice tumor model. Our results showed that the USP8 inhibitor combined with anti-PD-L1 treatment significantly elevated the expression of GzmB and reduced the expression of TIM3 on infiltrated CD8$^+$ T cells (Fig. 6i, j). In addition, significant upregulation of PD-L1, p-p65 and MHC-I were also observed in tumor tissues treated with the USP8 inhibitor or combined treatment compared with control treatment (Supplementary Fig. 6i–o). These results suggest that the USP8 inhibitor treatment might reprogram an inflamed TME evidenced by upregulation of PD-L1 and activation of NF-κB to promote the gene expression of MHC-I presenting pathway, which enhances the tumor-infiltrating CD8$^+$ cytotoxic T cells to enable the PD-1/PD-L1 blockade in vivo.

In keeping with the observations of USP8 inhibitor treatment in vivo, genetic depletion of *Usp8* also sensitized CT26 tumors to anti-PD-L1 immunotherapy in immunocompetent BALB/c mice

(Fig. 6k, Supplementary Fig. 6p, q). Moreover, PD-L1 expression, MHC-I and the infiltrated CD8$^+$ cytotoxic T cells were significantly upregulated in *Usp8*-deficient tumor tissues treated with control IgG or anti-PD-L1 antibody (Fig. 6l, m and Supplementary Fig. 6r, s). Although these results demonstrate that tumor-specific genetic depletion of USP8 could significantly enhance the therapeutic efficacy of PD-L1 blockade, the systemically using USP8 inhibitor treatment may also affect the function of other cells including immune cells in vivo.

Our results above suggested that USP8 regulated the efficacy of anti-PD-1/PD-L1 immunotherapy largely through two arms: the TRAF6-PD-L1 axis and the TRAF6-NF-κB-MHC-I pathway, to shape the tumor microenvironment (Fig. 6n). In keeping with the crucial role of TRAF6 in controlling both arms, depletion of *Traf6* abolished the sg*Usp8*-driven upregulation of PD-L1 protein abundance and the MHC-I expression in CT26 cells (Supplementary Fig. 7a–e). Furthermore, *Traf6* deficiency almost abolished the *Usp8*-deficient CT26 tumors sensitization to anti-PD-L1 immunotherapy in syngeneic mouse tumor model (Supplementary Fig. 7f–h). Expression of PD-L1 and MHC-I on surface of tumor cells, and tumor-infiltrating CD8$^+$ cytotoxic T cells were significantly decreased in *Usp8/Traf6* double KO CT26 tumors compared with *Usp8*-deficient CT26 tumors (Supplementary Fig. 7i–l). These results suggest that *Traf6* deficiency largely alleviates *Usp8* KO-driven anti-tumor effect via altering the TME in vivo.

To further dissect the role of each arm in regulating the tumor immunotherapy, we applied the *Pd-l1*-deficient CT26 syngeneic mouse tumor model to block the TRAF6-PD-L1 arm. Our results demonstrated that although USP8 inhibitor treatment could significantly suppress the *Pd-l1*-deficient CT26 tumor growth, there was no further additive effect when combined with anti-PD-L1 treatment (Supplementary Fig. 7m). However, p-p65, MHC-I and tumor-infiltrating CD8$^+$ cytotoxic T cells were significantly elevated in *Pd-l1*-deficient CT26 tumor tissues treated with the USP8 inhibitor alone or combination compared with the control group (Supplementary Fig. 7n–r). These results support the notion that the upregulation of PD-L1 protein abundance by USP8 inhibition is required for enhancing the therapeutic effect of anti-PD-L1 immunotherapy.

As depletion of *p65* significantly decreased the MHC-I, but not PD-L1 in *Usp8*-deficient CT26 cells (Fig. 5l–o and Supplementary Fig. 5q, r), which mimics blocking the TRAF6-NF-κB-MHC-I arm. Thus, we examined whether depletion of *p65* could compromise the *Usp8*-deficient CT26 tumors response to anti-PD-L1 immunotherapy in vivo. Our results demonstrated that *Usp8*-deficient CT26 tumors sensitized to the PD-L1 blockade compared with *Usp8/p65* double deficient CT26 tumors upon the anti-PD-L1 immunotherapy (Supplementary Fig. 7s–u), which might be due to *p65* deficiency abolishing the sg*Usp8*-driven upregulation of MHC-I and infiltrated CD8$^+$ cytotoxic T cells (Supplementary Fig. 7v–y). These results indicated that the NF-κB activation by USP8 inhibition is also needed to sensitize tumors to the anti-PD-L1 immunotherapy in vivo. Taken

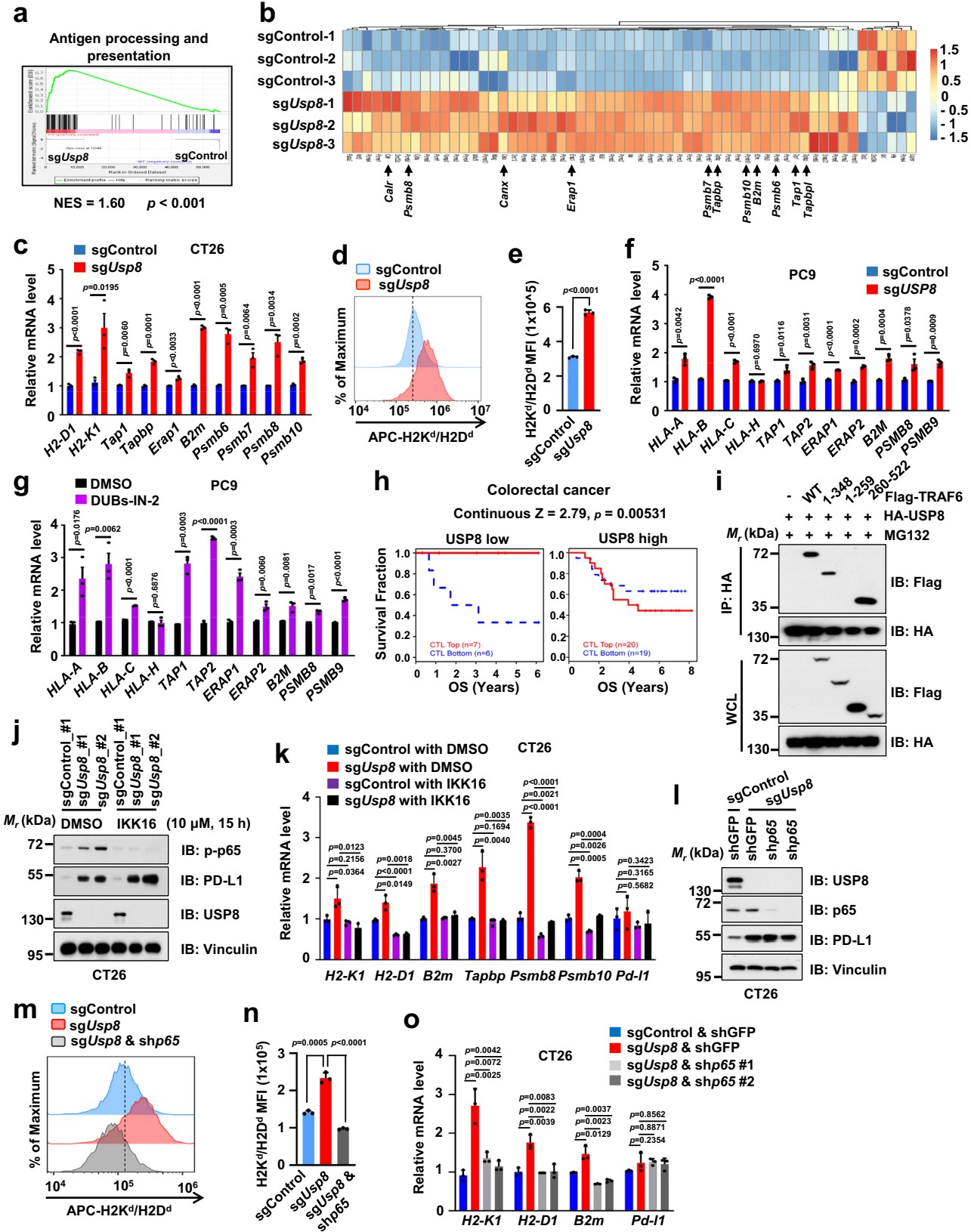

together, these results reveal a molecular mechanism and potential strategy of combination therapy of USP8 inhibitor plus anti-PD-1/PD-L1 antibody to enhance the efficacy of anti-tumor therapy largely through reprograming an inflamed TME.

## Discussion

Ubiquitination/de-ubiquitination is a reversible process that is controlled by ubiquitin E3 ligases and DUBs. DUB dysregulation is involved in many human diseases including cancer, which

**Fig. 5 Inhibition of USP8 elevates the antigen presentation pathway largely through activating the TRAF6-NF-κB signaling. a** Gene-set enrichment analysis (GSEA) for MHC-I-dependent antigen processing and presentation pathway genes in sgUsp8 versus sgControl cells. $n = 3$ biologically independent samples per group. $p$ values are calculated using Kolmogorov–Smirnov tests. NES: normalized enrichment score. **b** Heatmap showing differential expression of genes in Fig. 5a. **c** mRNA levels of indicated genes from sgUsp8 or sgControl CT26 cells were analyzed using RT-qPCR. **d, e** Cell surface H2K$^d$/H2D$^d$ on sgUsp8 or sgControl CT26 cells was analyzed by flow cytometry. **f** mRNA levels of indicated genes from sgControl or sgUsp8 PC9 cells were analyzed using RT-qPCR. **g** mRNA levels of indicated genes from PC9 cells treated with DMSO or DUBs-IN-2 (2 μM) for 24 h were analyzed using RT-qPCR. **h** The association between cytotoxic T lymphocyte level (CTL) and overall survival (OS) for colorectal cancer patients (GSE71187 cohort) under the condition of USP8 high or low expression was analyzed using Kaplan–Meier curves by the Tumor Immune Dysfunction and Exclusion (TIDE) algorithm. Two-sided Wald-test. **i** Immunoblot (IB) analysis of whole-cell lysates (WCL) and anti-HA IPs from 293 T cells co-transfected with indicated constructs. Cells were treated with 10 μM MG132 for 12 h. Three independent experiments were conducted. **j** IB analysis of WCL derived from sgControl or sgUsp8 CT26 cells treated with indicated inhibitors of NF-κB (IKK16, 10 μM) for 15 h. Three independent experiments were conducted. **k** mRNA levels of indicated genes from sgControl or sgUsp8 CT26 treating with DMSO or 10 μM IKK16 for 15 h. **l-n** IB analysis of WCL derived from sgControl or sgUsp8 CT26 cells infected with indicated lentiviral shGFP or shp65 (**l**). Cell surface H2K$^d$/H2D$^d$ was analyzed by flow cytometry (**m, n**). **o** mRNA levels of indicated genes from sgControl or sgUsp8 CT26 cells infected with indicated lentiviral shGFP or shp65. For **c, e–g, k, n**, and **o** data were presented as mean ± S.D.; $n = 3$ biologically independent samples; Two-sided $t$-test. The relevant raw data and uncropped dots are provided as a Source Data file.

highlights that DUBs are potential targets for cancer therapy[21,22]. USP8 is one of the DUBs and is frequently overexpressed in multiple types of human cancer[25–27]. Moreover, somatic *USP8* gain-of-function mutations are common in ACTH-secreting pituitary adenoma causing Cushing's disease[28,29]. Importantly, USP8 has been identified as an immunomodulatory DUB as T-cell-specific *Usp8* deficiency disrupts regulatory T-cell functions, leading to recruiting abundant CD8$^+$ γδT cells in colons and resulting in inflammatory bowel disease in mice[30]. These reports indicate that overexpression or gain-of-function mutation of USP8 may promote tumorigenesis through supporting regulatory T-cell functions and suppressing CD8$^+$ T-cell functions, leading to cancer immune evasion. However, whether targeting USP8 can enhance anti-tumor immunity has not been reported.

Here, we found that USP8 negatively regulates PD-L1 protein abundance largely through removing K63-linked ubiquitination on PD-L1, leading to increased K48-linked ubiquitination and degradation of PD-L1 (Figs. 1 and 2). Importantly, inhibiting USP8 by the pharmacological inhibitor or genetic depletion not only significantly elevates PD-L1, but also triggers innate and adaptive immune response (Figs. 1, 4, and 5). Through the bioinformatic analysis, we found that colon and lung cancer patients with low USP8 expression had significant upregulation of a panel of MHC-I pathway-related genes, which indicates a better survival when accompanied with a higher level of CTLs infiltration (Fig. 5h and Supplementary Fig. 5a, b).

In contrast to previous reports that several ubiquitin E3 ligases including SPOP and β-TRCP negatively regulate PD-L1 stability largely through promoting the poly-ubiquitination and degradation of PD-L1[17,18]. Here, through screening the E3 ligase library we found that TRAF6 interacts with and positively regulates PD-L1 stability through promoting the K63-linked ubiquitination of PD-L1 in cancer cells (Fig. 3). Although it is well-characterized that TRAF6 is critical for the regulation of innate and adaptive immunity largely through its E3 ligase activity for promoting the K63-linked ubiquitination of key factors[49], whether TRAF6 is involved in regulating anti-tumor immunity remains obscure. Our results demonstrate that TRAF6 stabilizes PD-L1 and upregulates the MHC-I-dependent antigen presentation, suggesting that TRAF6 is directly involved in cancer immunotherapy. However, whether cancer patients with high TRAF6 expression are more sensitive to PD-1/PD-L1 blockade requires further investigation.

Lastly, our study identified a combined therapeutic strategy that the combination of USP8 inhibitor and PD-1/PD-L1 blockade could significantly reduce tumor growth and increase overall survival rate in different mouse tumor models (Fig. 6). Together, our study demonstrates that high expression of USP8

in tumors without treatment might inhibit TRAF6-mediated K63-linked ubiquitination of PD-L1 to induce low PD-L1 expression and suppress the immune response and the MHC-I-mediated antigen presentation through inhibiting the TRAF6-NF-κB signaling, leading to a non-inflamed TME and resistance to PD-1/PD-L1 blockade (Fig. 6n, left panel). However, inhibition of USP8 using the inhibitor DUBs-IN-2 can reverse this process to upregulate PD-L1 as well as trigger immune response and antigen presentation, reshaping an inflamed TME where anti-PD-1/PD-L1 immunotherapy can be more effective (Fig. 6n, right panel). Hence, our study not only provides a molecular insight but also reveals a potential therapeutic strategy that targeting the immunomodulatory deubiquitinase USP8 might enhance the efficacy of anti-PD-1/PD-L1 in treating human cancers.

## Methods

**Mouse model.** Animal studies were approved by the Institutional Animal Care and Use Committee of the Medical University of South Carolina (protocol number 2018-00500-1) or Wuhan University. MC38 or CT26 cells in PBS or DMEM were subcutaneously injected into the flank of 6-week-old C57BL/6 or BALB/c female mice, respectively (Jackson Laboratory). Kras$^{LSL-G12D/+}$Tp53$^{fl/fl}$ (KP) mice were kindly provided by the laboratory of Dr. Bo Zhong (Wuhan University). Mice were maintained in Specific Pathogen Free (SPF) animal facility (68-71.6 °F temperature and 50%-60% humidity). The dark/light cycle animal rooms: 12 h of light and 12 h of dark. All mice experiments were conducted following animal ethical regulations and the study protocol.

**Cell culture, transfection, and generating stable cell lines.** HEK293T, H460, PC9, A375, B16-F10, CT26, MC38, and U2OS cells were cultured in DMEM (Hyclone) medium supplemented with 10% FBS (Gibco), 100 units of *penicillin*, and 100 mg/ml *streptomycin*. DU145 cells were maintained in RPMI-1640 (Hyclone) medium supplemented with 10% FBS, 100 units of *penicillin*, and 100 mg/ml *streptomycin*. All cells were regularly authenticated by short tandem repeats analysis and tested for the absence of Mycoplasma contamination using MycoAlert (Lonza).

Cells with 60–80% confluence were transfected with indicated constructs using Lipofectamine 2000 (Life Technologies) or Polyethylenimine (PEI, Polysciences) in OptiMEM medium (Gibco) according to the manufacturer's instructions. Thirty-six hours post-transfection, cells were harvested and subjected to various assays. For gene knockdown or knockout, lentiviral constructs (pLKO.1 for shRNAs and pLenti-V2 for sgRNAs) were transfected into 293T cells together with helper plasmids (pVSVG and pD8.9) using PEI or Lipofectamine 2000. Viral supernatants were collected at 36 and 48 h post-transfection. Cells with around 50% confluence were infected with viral supernatants supplemented with 4 μg/ml polybrene (Sigma). Following viral infection, cells were selected in the presence of puromycin (1 μg/ml or 8 μg/ml) for at least 3 days to generate stable cell lines.

**Plasmids.** pcDNA3-PD-L1 and pcDNA3-HA-PD-L1 have been described previously[17]. Flag-USP8 (WT and mutants: amino acid 1–313, 314–714, 715–1118, and C786A) and Flag-TRAF6 (WT and mutants: amino acid 1–259, 1–348, 260–499, and C70A) were amplified and cloned into pcDNA3-Flag vector. USP8 and TRAF6 were amplified and cloned into pLenti-HA vector. HA-tagged TRAF1,

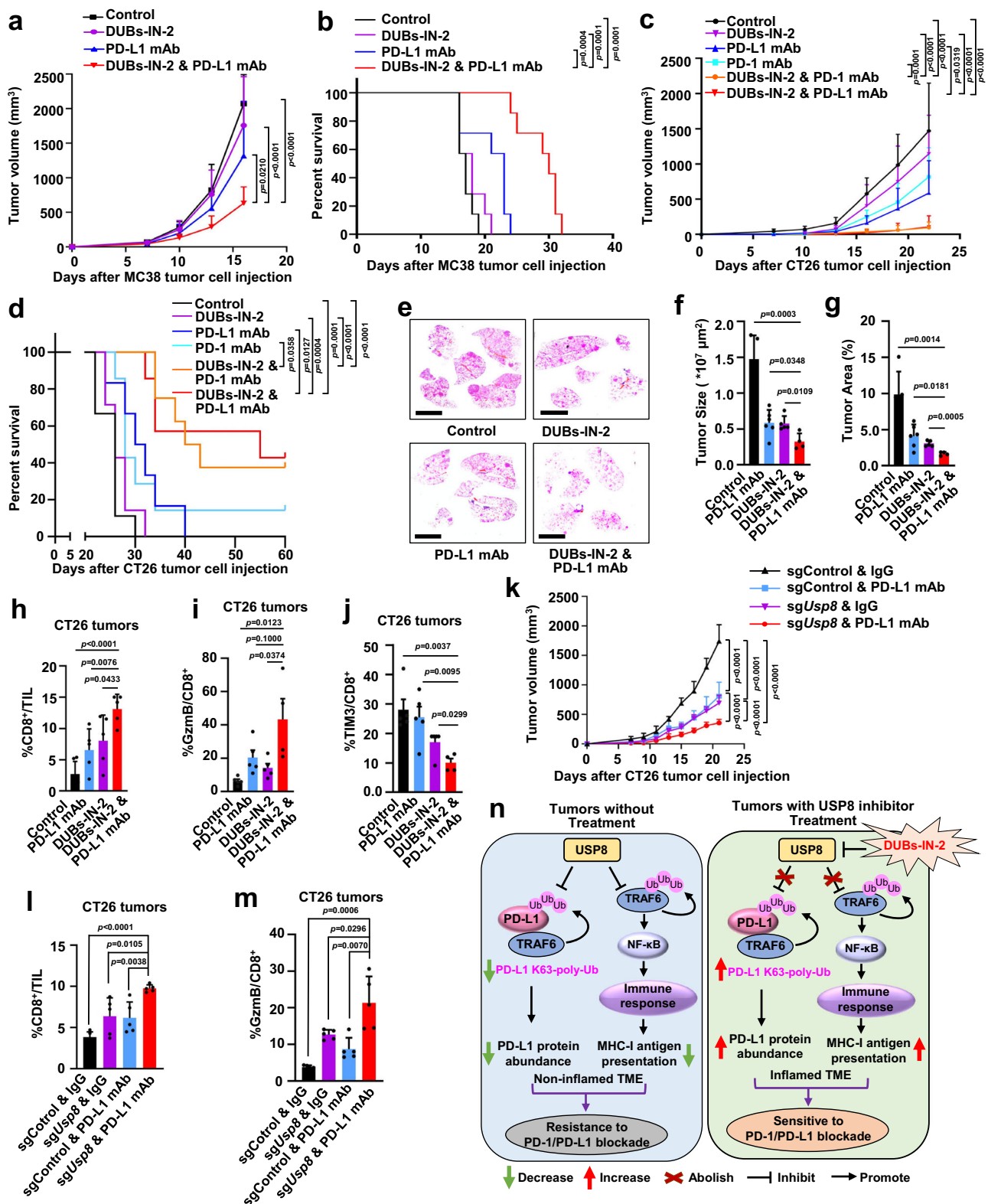

TRAF2, TRAF3, and Skp2 have been described previously[50]. Flag-TRAF2, TRAF3, TRAF4, TRAF5, TRAF7, USP2, USP7, USP10, USP13, USP14, and USP20, TRAF2 were purchased from Origene. PD-L1-Luciferase constructs and E3 ligase library for luciferase screening were provided by the Laboratory of Dr. Hong-Bing Shu. His-Ub and mutants have been described previously[51]. shRNAs targeting USP8 were purchased from Open Biosystems. shRNA sequences for mouse Usp8: 5′-TTGTAAGCATTAGATGTGAGG-3′(#744); 5′-TAGCATTGGTTGTAAACT GCG-3′(#745). shRNA sequences for mouse p65: 5′-ATGGATTCATTACAGC TTAAT-3′(#1); 5′-CGGATTGAGGAGAAACGTAAA-3′(#2). sgRNAs for human TRAF6: 5′-GTAACAAAAGATGATAGTGT-3′(#5); 5′-TGGGTGGAACTGC

CAGCACG-3′(#6). sgRNA for human USP8 was 5′-GATTTTACTTATCCCTCA TTGG-3′. sgRNAs for mouse Usp8: 5′-TGAAGAAAAGGACAGACGGG-3′(#1); 5′-GGTCTTTTAGTGAAGAACTG-3′(#2). sgRNAs for mouse Traf6: 5′-CCTCT CCAGCTCCTTCATGG-3′(#4); 5′-GCAGTATTTCATTGTCAACT-3′(#5). Control sgRNA sequence: 5′-CTTGTTGCGTATACGAGACT-3′(#1); 5′-CGCTTCC GCGGCCCGTTCAA-3′(#2).

**Compounds.** DUBs-IN-2 (HY-50737A), ML323 (HY-17543), ML364 (HY-100900), P22077 (HY-13865), Spautin-1 (HY-12990), IU1 (HY-13817), LDN-

**Fig. 6 The combination of USP8 inhibitor with PD-1/PD-L1 blockade significantly suppresses tumor growth in vivo. a, b** Tumor growth or Kaplan–Meier survival curves for C57BL/6 bearing MC38 tumors with indicated treatments. $n = 7$ mice/group. log-rank test (**b**). mAb: monoclonal antibody. **c, d** Tumor growth or Kaplan–Meier survival curves for BALB/c mice bearing CT26 tumors with indicated treatments. $n = 9$ (control), 7 (DUBs-IN-2), 6 (PD-L1 mAb), 7 (PD-1 mAb), 8 (PD-1 mAb plus DUBs-IN-2) or 7 (PD-L1 mAb plus DUBs-IN-2) mice. log-rank test (**d**). **e–g** Representative images of HE staining (**e**), tumor sizes (**f**), or tumor areas (**g**) in tumor-burdened lungs of KP mice were analyzed. Scale bars represent 5 mm. $n = 5, 6, 5,$ or 4 mice/group. **h** Quantification of CD8[+] T cells represented as percentage of tumor-infiltrating lymphocytes (TIL) in CT26 tumors after indicated treatments. $n = 6, 5, 5,$ or 5 mice/group. **i, j** Quantification of Granzyme B (GzmB) (**i**) or TIM3 (**j**) represented as percentage on CD8[+] TILs in CT26 tumors after indicated treatments. $n = 5, 5, 5,$ or 4 mice/group. **k** Tumor growth of sgControl or sg*Usp8* CT26 cells in BALB/c mice with indicated treatments. $n = 5$ mice/group. **l** Quantification of CD8[+] T cells represented as percentage of TIL in sgControl or sg*Usp8* CT26 tumors after indicated treatments. $n = 5$ mice/group. **m** Quantification of GzmB represented as percentage on tumor-infiltrating CD8[+] T cells in sgControl or sg*Usp8* CT26 tumors after indicated treatments. $n = 5$ mice/group. **n** A working model for targeting USP8 sensitizes tumors to PD-1/PD-L1 blockade. USP8 downregulates PD-L1 and MHC-I-mediated antigen-presenting, leading to non-inflamed TME and resistance to PD-1/PD-L1 blockade (left panel). However, inhibition of USP8 by DUBs-IN-2 upregulates PD-L1 and antigen-presenting, setting up an inflamed TME and sensitive to anti-PD-1/PD-L1 immunotherapy (right panel). Ub: ubiquitin. TME: tumor microenvironment. For **a, c, k** data were presented as mean ± S.D.; two-way ANOVA test. For **f, g, h–j, l, m** data were presented as mean ± S.D.; two-sided *t*-test. Source data are provided as a Source Data file.

57444 (HY-18637), PR-619 (HY-13814), BAY 11-7082 (HY-13453), MF-094 (HY-112438), IKK 16 (HY-13687) were purchased from MedChemExpress. TCID (S7140) was purchased from Selleck. MG132 (BML-PI102-0005) was purchased from Enzo life science. Cycloheximide (C7698-5G) was purchased from Sigma. Anti-mouse PD-1 (clone 29F.1A12)/PD-L1 (clone 10 F.9G2) for mice treatment antibodies were provided by the Laboratory of Dr. Gordon J. Freeman.

**Immunoblot and immunoprecipitation.** Cells were harvested and lysed in EBC buffer (50 mM Tris pH 7.5, 120 mM NaCl, 0.5% NP-40) containing protease inhibitors (Protease inhibitor cocktail 100x in DMSO, Cat. No. B14002, Bimake) and phosphatase inhibitors (phosphatase inhibitor cocktail, Cat. No. B15002, Bimake). Total protein concentrations were measured by the spectrophotometer of Thermo Multiskan FC using the BCA Protein Quantification Kit. For immuno-precipitation assays, 1–2 mg whole-cell lysate protein were incubated with bead-conjugated anti-Flag/anti-HA or other appropriate antibodies (2 μg) in a rotating incubator overnight at 4 °C. Immuno-complexes were washed four times with NETN buffer (20 mM Tris, pH 8.0, 100 mM NaCl, 1 mM EDTA and 0.5% NP-40). Both lysates and immunoprecipitates were resolved by SDS-PAGE and immuno-blotted with indicated antibodies through ECL chemiluminescent Detection Reagent.

The following is the information about antibodies used in immunoblot and immunoprecipitation: Anti-PD-L1 (E1L3N) rabbit mAb (13684), anti-TRAF6 (D21G3) rabbit mAb (8028), anti-B7-H3 (D9M2L) rabbit mAb (14058), anti-Phospho-p65 (93H1) rabbit mAb (3033), anti-LC3B (E7X4S) rabbit mAb (43566), anti-K63-linkage Specific Polyubiquitin (D7A11) rabbit mAb (5621) and anti-K48-linkage Specific Polyubiquitin rabbit pAb (4289) were purchased from Cell Signaling Technology and diluted 1:2000. Anti-PD-L1 [EPR20529] mouse mAb (ab213480) was purchased from Abcam and diluted 1:2000. Anti-TRAF6 (H-274) rabbit mAb (sc-7221), Anti-USP8 (UBPY) (E-1) mouse mAb (sc-376130), anti-JNK (D-2) mouse mAb (sc-7345), and anti-RelA (5G8) mouse mAb (sc-81622) were purchased from Santa Cruz Biotechnology and diluted 1:1000. Anti-Vinculin (VIN-11-5) mouse mAb (V4505), anti-CMTM6 rabbit pAb (SAB2701009), anti-Flag rabbit pAb (F7425), anti-Flag M2 mouse mAb (F3165), anti-HA Agarose (A2095), anti-Flag M2 affinity gel (A2220), anti-HA rabbit pAb (H6908), peroxidase-conjugated anti-mouse secondary antibody (A-4416), and peroxidase-conjugated anti-rabbit secondary antibody (A-4914) were purchased from Sigma-Aldrich and diluted 1:5000. Anti-Purified anti-HA.11 Epitope Tag (16B12) mAb (MMS-101P) were purchased from Biolegend and diluted 1:5000. Anti-human PD-L1 (clone 29 E.12B1) for immunoprecipitation was provided by the Laboratory of Dr. Gordon J. Freeman.

**Reverse transcription quantitative PCR (RT-qPCR) analysis.** Total RNAs were extracted using the TRIzol reagent (Invitrogen), and reverse transcription reactions were performed using the PrimeScript RT reagent kit (TaKaRa, Cat. No.RR470A) with a mix of random 6 mers and oligo(dT) primers. After mixing well generated cDNA templates with primers/probes and PerfectStart Green qPCR SuperMix (Transgen, Cat. No. AQ601), RT-qPCR was performed with the Bio-Rad CFX Connect Real-Time PCR Detection System (Bio-Rad). The housekeeping gene, GAPDH, was used as a loading control. Primers were listed in Supplementary Table 1.

**Analysis of membrane PD-L1 by the flow cytometry.** Cells were washed once in PBS and stained using the APC or PE conjugated PD-L1 for 30 min at 4 °C. After staining, samples were fixed for 30 min at 4°C using the eBioscience™ Fixation/Permeabilization kit. After washing once in PBS, cells were analyzed and data were acquired on Beckman CYTOFLEX and Beckman CytExpert Software 2.3. Results were analyzed by the software FlowJo and the GraphPad.

**Protein half-life analysis.** Cells undergoing *Usp8* depletion or *TRAF6* over-expression were subjected to cycloheximide (200 or 400 μg/ml, Sigma) treatment for indicated time courses. Cells were harvested for immunoblot analysis using indicated antibodies. PD-L1 protein band densities were quantified by ImageJ software and normalized to vinculin.

**In vivo ubiquitination assays.** Cells with 80% confluence were transfected with His-ubiquitin (His-Ub) and desired constructs. Thirty-six hours after transfection, cells were treated with 10 μM MG132 overnight and lysed in denaturing buffer A (6 M guanidine-HCl, 0.1 M $Na_2HPO_4/NaH_2PO_4$, and 10 mM imidazole [pH 8.0]). After sonication, cell lysates were incubated with nickel-nitrilotriacetic acid (Ni-NTA) beads (QIAGEN) for 3 h at room temperature. Subsequently, Ni-NTA beads were washed twice with buffer A, twice with buffer A/TI (vol: vol = 1:3), and once with buffer TI (25 mM Tris-HCl and 20 mM imidazole [pH 6.8]). In all, 30 μl 2x protein loading buffer were added into Ni-NTA beads and boiled for 10 min. Pull-down proteins were resolved by SDS-PAGE for immunoblotting using indicated antibodies.

**Dual-luciferase assays for screening E3 ligases.** HEK293 cells were seeded on 24-well plates and transfected when cell confluences got up to 70%. Each trans-fection was composed of 0.01 μg PD-L1-Firefly luciferase, 0.1 μg one of indicated E3 ligase as well as 0.01 μg pRL-TK (Renilla luciferase) reporter plasmid as the internal control. Twenty-four hours after transfection, cells were harvested and luciferase activity was measured using a Dual-Luciferase Assay Kit according to manufacturer's instructions (Promega). The relative firefly luciferase activity was normalized to renilla luciferase activity and fold-change was normalized to the control value of pCMV6.

**RNA sequencing (RNA-seq) and bioinformatic analysis.** Total RNAs were isolated from $1 \times 10^6$ either the *Usp8*-WT or KO CT26 cells by using TRIzol Reagent following the manufacturer's instructions (Invitrogen). RNA libraries were constructed by the Beijing Genomics Institute and sequenced on the BGISEQ platform with paired-end reads (150-bp read length). For analysis of RNA-seq results, RNA-seq reads quality was evaluated using FastQC (v0.11.9, https://www.bioinformatics.babraham.ac.uk/projects/fastqc/) and aligned to the mouse genome GRCm38 by HISAT2 (v2.2.1, https://daehwankimlab.github.io/hisat2/). FeatureCounts (v2.0.1, http://subread.sourceforge.net/) was used to quantitate the transcriptome using the GTF annotation files. Differential analyses were performed to the count files using the R packages DESeq2 (v1.28.1, https://bioconductor.org/packages/release/bioc/html/DESeq2.html), following standard normalization procedures. The differentially expressed genes (DEGs) were identified with adjusted *p*-value < 0.05 and absolute log$_2$ fold-change > 1 and plotted with R packages ggplot2 (v3.3.2, https://cran.r-project.org/web/packages/ggplot2/index.html). Heatmaps were generated using pheatmap package (v1.0.12, https://cran.r-project.org/web/packages/pheatmap/index.html) in R (v4.0.2, https://cran.r-project.org). Gene Ontology (GO) enrichment analysis was performed by the R packages cluster-Profiler (v3.16.1, https://bioconductor.org/packages/3.13/bioc/html/clusterProfiler.html). Gene-set enrichment analysis was performed using the GSEA software (v4.1.0, https://www.gsea-msigdb.org/gsea/index.jsp). The RNA-seq data from *Usp8* WT and KO CT26 cells will be deposited in the Gene Expression Omnibus (GEO) database.

**Transcripts and survival analyses.** The data for TRAF6 and USP8 expression and survival of cancer patients were generated using the TIDE tool (http://tide.dfci.harvard.edu)[44]. mRNA expression *z*-score data of LUAD and COAD in TCGA cohort are downloaded from cBioportal[52].

**Immunohistochemistry (IHC), digital pathology, and scoring system**. Human lung tissue microarrays containing 73 of lung squamous cancer patient tissues (OD-CT-RsLug01-009) were purchased from Shanghai Outdo Biotech, China. This study was conducted strictly based on the guidelines including obtained informed consent from all participants by the Medical Ethics Committee of Shanghai Outdo Biotech (Project Number: YB M-05-02). IHC methods and processes were reported previously[17]. Briefly, tissues were cut into 4-μm sections. After deparaffinizing and rehydrating, sections were boiled in 0.01 M citric acid buffer solution (pH 6.0) for 1.5 min at high pressure. Subsequently, samples were incubated with 3% hydrogen superoxide for 20 min to quench endogenous peroxidase activity, and 10% goat serum was used to block non-specific binding. Samples were incubated with anti-human PD-L1 and TRAF6 or USP8 antibodies or isotype-matched IgG controls overnight at 4 °C. A positive slide was set at each experiment. Next, a secondary biotinylated immunoglobulin G antibody solution and an avidin-biotin peroxidase reagent were added onto slides. After washing with phosphate buffer saline, 3,3′-diaminobenzidine tetrachloride was added to the sections, followed by counter-staining with Mayer's hematoxylin. The process for lungs from KP mice to stain with anti-mouse PD-L1 or anti-mouse H2K$^b$ antibodies were similar to IHC methods and processes that described above.

Anti-TRAF6 rabbit pAb (A16991) was purchased from ABclonal and diluted 1:600. Anti-mouse PD-L1 (10F.9G2) mAb (124302), anti-mouse H2K$^b$ (AF6-88.5) mAb (116502) were purchased from Biolegend and diluted 1:100. Anti-human PD-L1 (E1L3N) rabbit mAb (13684) were purchase from Cell Signaling Technology and diluted 1:100. Anti-USP8 (UBPY) (E-1) mouse mAb (sc-376130) were purchase from Santa Cruz Biotechnology and diluted 1:100.

Immunohistochemical staining was scanned using an Aperio ScanScope CS whole slice scanner (Vista, CA, USA) with background subtraction as previously described[53]. The membrane, cytoplasm, or pixel immunohistochemical staining was quantified using Aperio Quantification software. The histoscores of the membrane and cytoplasm staining quantification were assessed according to the formula: $(3 + percent cells) \times 3 + (2 + percent cells) \times 2 + (1 + percent cells) \times 1$. The formula total intensity/total cell number was used to assess the histoscore of pixel quantification.

**In vivo experimental therapy in syngeneic mouse tumor models**. Animal studies were approved by the Institutional Animal Care and Use Committee of the Medical University of South Carolina (IACUC; protocol number 2018-00500-1) or Wuhan University. $2 \times 10^5$ MC38 or $1 \times 10^5$ CT26 cells in 200 μl PBS were subcutaneously injected into the flank of 6-week-old C57BL/6 or BALB/c female mice (Jackson Laboratory), respectively. On the day of 7 or 10 after tumor cells implantation, tumor sizes were measured every 3 days by caliper and tumor volumes were calculated by the formula: $length \times width^2 \times 0.5$. The mice were euthanized when the tumor size is bigger than 20 mm of the diameter or tumor volume reaches 2000 mm$^3$ and deemed as death. Tumor-bearing mice were pooled and randomly divided into the following groups: (1) control; (2) USP8 inhibitor (DUBs-IN-2); (3) anti-PD-L1 antibody (clone 10 F.9G2); (4) anti-PD-1 antibody (clone 29F.1A12); (5) anti-PD-L1 antibody plus USP8 inhibitor or (6) anti-PD-1 antibody plus USP8 inhibitor. All treatments were conducted by intraperitoneal injection. As shown in supplementary Fig. 6a, c, anti-PD-L1 or PD-1 antibody was applied every 3 days. The USP8 inhibitor treatment was given with a dosage of 3 mg/kg of mouse body weight daily with a break every 6 days.

In all, $1 \times 10^6$ sgControl or sgUsp8 CT26 cells in 100 μl DMEM were subcutaneously injected into the flank of 6-week-old BALB/c female mice (GemParmatech), respectively. On the day of 7 after tumor cells implantation, tumor sizes were measured every 2 days by caliper. Tumor-bearing mice were pooled and randomly divided into the following groups: (1) CT26 sgControl & control IgG; (2) CT26 sgUsp8 & control IgG; (3) CT26 sgControl & anti-PD-L1 antibody (clone 10F.9G2) or (4) CT26 sgUsp8 & anti-PD-L1 antibody. The anti-PD-L1 or control IgG antibodies treatment were given with a dosage of 100 μg/mouse every 3 days for five times.

In all, $5 \times 10^6$ sgControl, sgUsp8 or sgUsp8 & sgTraf6 CT26 cells in 150 μl DMEM were subcutaneously injected into the flank of 6-week-old BALB/c female mice (GemParmatech), respectively. On the day of 7 after tumor cells implantation, tumor sizes were measured every 2 days by caliper. The anti-PD-L1 antibody treatment was applied to every group with a dosage of 100 μg/mouse every 3 days for three times.

In all, $1 \times 10^5$ sgPd-l1 CT26 cells in 200 μl PBS were subcutaneously injected into the flank of 6-week-old BALB/c female mice, respectively (Jackson Laboratory). On the day of 12 after tumor cells implantation, tumor sizes were measured every 3 days by caliper Tumor-bearing mice were pooled and randomly divided into the following groups: (1) control; (2) USP8 inhibitor (DUBs-IN-2); (3) anti-PD-L1 antibody (clone 10 F.9G2) or (4) anti-PD-L1 antibody plus USP8 inhibitor. The anti-PD-L1 antibody was applied every 3 days. The USP8 inhibitor treatment was given with a dosage of 3 mg/kg of mouse body weight daily with a break every 6 days.

In all, $1 \times 10^6$ sgUsp8 or sgUsp8 & shp65 CT26 cells in 100 μl DMEM were subcutaneously injected into the flank of 6-week-old BALB/c female mice (GemParmatech), respectively. On the day of 7 after tumor cells implantation, tumor sizes were measured every 2 days by caliper. The Anti-PD-L1 antibody treatment was applied to every group with a dosage of 100 μg/mouse every 3 days for four times.

For survival studies, mice were monitored and measured for tumor volumes twice a week after initial injections. Mice were sacrificed when tumor volume exceeded 2000 mm$^3$ or tumor had ulcers with diameter reached 1 cm. Statistical analysis was performed using the GraphPad Prism 8.0 software. Kaplan–Meier survival curves and corresponding log-rank (Mantel-Cox) tests were used to evaluate the statistical differences between groups in survival studies. There is a significant difference when the $P < 0.05$.

**Induction of tumorigenesis, treatment, and Hematoxylin-Eosin (H&E) staining in Kras$^{LSL-G12D/+}$Tp53$^{fl/fl}$ (KP) mouse model**. Kras$^{LSL-G12D/+}$Tp53$^{fl/fl}$ (KP) mice were kindly provided by the laboratory of Dr. Bo Zhong (Wuhan University). These mice were bred for maintenance and experiments. Induction of tumorigenesis was performed as previously described[54]. Seven to 8-week-old were anesthetized with 1% sodium pentobarbital ($w/v = 1:7$), followed by intranasal instillation of Adenovirus-Cre (Ad-Cre, $1–2 \times 10^6$ pfu in 50 μl PBS per mouse, Obio Technology, Shanghai, China). At fifth week after tumor induction, mice were treated with the USP8 inhibitor/anti-PD-L1 alone or combined as shown in supplementary Fig. 6e, anti-PD-L1 antibody was applied every 3 days. The USP8 inhibitor treatment was given with a dosage of 1 mg/kg of mouse body weight daily with a break every 6 days. After accomplishing the treatment, mice were euthanized for the Bronchoalveolar Lavage Fluid (BALF). Subsequently, lungs from mice were fixed in 4% paraformaldehyde for 4 h at room temperature and were left in 75%, 95%, 100% EtOH for 2 h in every concentration before soaking in xylene for 4 h. After dehydration, lungs were treated in liquid paraffin until the paraffin embedding. The paraffin blocks were sectioned (5 μm) for H&E staining (Beyotime Biotech). We used the Aperio VERSA 8 (Leica) multifunctional scanner to get image and analysis.

**Tumor infiltrated immune cells isolation and flow cytometry analysis**. Tumor infiltrated immune cells were performed as previously described[55]. Briefly, tumors separated from the mice were minced by using two single-edged razor blades. The rubber plunger of a syringe would be used to mesh tissues through the 70 μm cell strainer in 100 mm dish. The cell suspension would be passed through another 70 μm cell strainer to 50 ml conical tube. The volume cell suspension would be adjusted to 30 ml with RPMI-1640 media at room temperature. 10 ml of Ficoll-Paque PREMIUM 1.084 would be slowly released to the bottom of 50 ml conical tube, which contained cell suspension. The solution was centrifuged at $1025 \times g$ for 20 min at 20 °C. We discarded the upper layer of media and transferred the layer of mononuclear cells to another 50 ml conical tube. Next, we used the complete RPMI-1640 to wash the mononuclear cells twice at 650 g for 10 min every time. For membrane staining, we used the PBS to suspend the cells and stained with antibodies for 15 min in the dark and then detected by flow cytometry. Cells were analyzed and data were acquired on BD Fortessa X-20 and FACSDiva 7 software following the exemplifying gating strategy for flow cytometry analysis (Supplementary Fig. 8). The data were processed using FlowJo software.

The following is the information about antibodies used in flow cytometry analysis: APC anti-human CD274 (10F.9G2) mAb (124311), PE anti-mouse CD274 (10F.9G2) mAb (124307), TCR-(H57-597) mAb (109227), Alexa Fluor®-700-CD8a (53-6.7) mAb (100730), APC/Cy7-CD4 (RM4-5) mAb (100526), Brilliant Violet 421™-CD45R/B220 (RA3-6B2) mAb (103251), APC-Cy7 CD11c (N418) mAb (117323), Perp-Cy5.5-CD45 (30-F11) mAb (103131), PE/Cy7-Granzyme B (QA16A02) mAb (372213), FITC-IFNγ(XMG1.2) mAb (505806), PE/Cy7-CD3666 (TIM3) (RMT3-23) mAb (119715), FITC-H2K$^b$ (AF6-88.5) mAb (116506), APC-H2K$^d$/H2D$^d$ (34-1-2 S) mAb (114713) were purchase from Biolegend and diluted 1:50. FITC-anti-mouse IA (MHC-II) (AF6-120.1) mAb (562011), PE-Cy7 CD11b (M1/70) mAb (561098), BV711-F4/80 (T45-2342) mAb (565612) were purchased from BD biosciences and diluted 1:100. Phospho-p65 (93H1) mAb (5733s) were purchase from Cell Signaling Technology and diluted 1:100.

**Statistical analysis**. The quantitative data are presented as mean ± S.D. of at least three independent experiments or biological replicates. Data analyses were carried out using GraphPad Prism 8.0 or Excel 2016 unless indicated otherwise. Statistical significances were analyzed using the unpaired, two-tailed Student's $t$-test and two-way ANOVA test. The correlation was analyzed using a Pearson correlation test. $P < 0.05$ were considered significant.

## Data availability

All data are available in the main text, Supplementary Information, or source data file. The RNA-seq data from sgControl and sgUsp8 CT26 cells generated in this study have been deposited in the Gene Expression Omnibus (GEO) database under the accession numbers GSE164558. The human cancer data (Fig. 5h and Supplementary Fig. 5h–j) were derived from GEO (Colorectal cancer GSE71187; Colorectal cancer GSE17536; Lung cancer GSE37745). Source data are provided with this paper.

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

## Acknowledgements

This work was supported by the National Natural Science Foundation of China (31970732), the startup funding from Wuhan University to J.Z., Key Research & Development Project of Hubei Province (2020BCA069) to C.X., Translational Medicine and Interdisciplinary Research Joint Fund of Zhongnan Hospital of Wuhan University (ZNJC201922) to C.X. and J.Z., Bristol-Myers Squibb-Melanoma Research Alliance (MRA) Young Investigator Award (821901) and NIH (R37CA251165) to H.W. and NCI P50CA101942 to G.J.F. We thank Dr. Bo Zhong in the Medical Research Institute at Wuhan University for generously providing *Kras*^G12D/+ *Tp53*^fl/fl (KP) mice; Dr. Minling Gao for critical reading and discussion of this manuscript. We also thank staff at the core facility of the Medical Research Institute at Wuhan University for their technical support.

## Author contributions

W.X. and X.G. performed most of the experiments with help from B.J., Y.G., L.Z., B.-L.X., C.H., Y.S., H.L., J.S., X.X., B.X., C.X., and G.C.; J.Z., H.W., W.X., and X.G. designed experiments. H.-B.S. and M.-M.H. supervised and performed luciferase assays for screening E3 ligases. X.B. and G.J.F. tested and provided PD-L1 antibodies for immunoprecipitation (clone 29E.12B1) and mouse therapeutic treatment (clone 10F.9G2). H.Z. supervised T.Z. for bioinformatic analysis. J.Z. and H.W. guided and supervised the project. J.Z. and W.X. wrote the manuscript. W.W., H.W., X.G., X.B., and G.J.F. edited the manuscript. All authors commented on the manuscript.

## Competing interests

G.J.F. has patents/pending royalties on the PD-1/PD-L1 pathway from Roche, Merck MSD, Bristol-Myers-Squibb, Merck KGA, Boehringer-Ingelheim, AstraZeneca, Dako, Leica, Mayo Clinic, and Novartis. G.J.F. has served on advisory boards for Roche, Bristol-Myers-Squibb, Xios, Origimed, Triursus, iTeos, NextPoint, IgM, and Jubilant. G.J.F. has equity in Nextpoint, Triursus, Xios, and IgM. W.W. is a co-founder and consultant for the ReKindle Therapeutics. Other authors declare no competing interests.
