## [Peer Review File · Nature Communications]

USP8 inhibition reshapes an inflamed tumor microenvironment that potentiates the immunotherapyREVIEWER COMMENTS

Reviewer #1 (Remarks to the Author):

Cancer immunotherapies targeting the PD-1/PD-L1 axis have demonstrated remarkable efficacy in a wide range of cancers. However, the molecular basis for resistance to PD-1/PD-L1 blockade is not fully understood. In this manuscript, Xiong and co-workers report a molecular mechanism by which USP8 regulates PD-L1 K63-linked ubiquitination and immune signalings to control anti-tumor immunity. They showed that inhibiting or depleting the deubiquitinase USP8 increases PD-L1 expression level through elevating TRAF6-mediated K63-linked ubiquitination of PD-L1, which may antagonize K48-linked ubiquitination and degradation. They also found that USP8 inhibition triggers innate immune responses through NF- κ B signaling. Taken together, their study indicates a potential combined therapeutic strategy that utilizes a USP8 inhibitor and PD1/PD-L1 blockade for enhancing efficacy of anti-tumor immunotherapies.

Overall, the authors provide clear evidence that USP8 regulates PD-L1 abundance and immune responses, placing USP8 as an important target for improving the efficacy of anti-PD-1/PD-L1 immunotherapies. The phenotypic data are compelling, including analysis of PD-L1 protein degradation, gene expression analysis using RNA-seq, and tumor growth in vivo. The relevance of USP8 inhibition in improving efficacy of anti-tumor immunotherapies will attract broad readers. My concerns relate to the proposed mechanism, e.g., USP8 is known to regulate the endosome-lysosomal pathway, but it is not clear from the data whether UPS8 regulates PD-L1 degradation through the proteasome or lysosome (see point 1 and 2). The authors should clarify the mechanism of PD-L1 degradation by adding more convincing data.

Major concerns:

1. USP8 is well known as a regulator of the endosomal sorting. Moreover, PD-L1 is a transmembrane protein, raising a possibility that USP8 may regulate degradation of PD-L1 through the endosome-lysosomal pathway. Although the involvement of USP8 in the regulating PD-L1 abundance is convincing (Fig. 1), it is not clear from the data presented whether the degradation is mediated by the proteasome or lysosome.
2. Related to the point 1, the proteasomal inhibitor MG132 was added to the most of cell-based ubiquitination analyses (Fig. 2 and Fig. 3). However, MG132-dependency of PD-L1 ubiquitination is not clear from the data. Can the ubiquitination of PD-L1 be detected only under proteasomal inhibition? Do lysosomal inhibitors such as bafilomycin A affect PD-L1 ubiquitination?
3. The authors use ubiquitin mutants (K48 or K63 only) to analyze PD-L1-conjugated ubiquitin chains. However, such experiments can sometimes be artificial, and the results should be validated using other experimental settings such as the use of linkage-specific antibodies (or, ideally, mass spectrometric analyses). For example, the authors claimed differential regulation of K48- and K63-linked ubiquitin chains by USP8 (Fig. 2I). These experiments should also be performed using linkage-specific antibodies, such as Apu2 and Apu3, to detect endogenous Ub-derived chains.
4. It has been reported that PD-L1 gene is induced by NF- κ B at the transcription level. Therefore, the presented data that USP8 also regulates NF- κ B signaling is a bit confusing to me. This may imply that USP8 regulates PD-L1 protein abundance, at least in part, through activating NF- κ B and PD-L1 gene induction. This apparent discrepancy should be clarified.
5. The authors show the direct interaction between USP8-PD-L1 (Fig. 2) and between TRAF6-PD-L1 (Fig. 3). They also show that USP8 and TRAF6 associate in cells (Fig. 5G). This raises a question whether USP8-PD-L1-TRAF6 form a ternary complex, or their interactions are antagonistic.

Minor point:

6. Is the regulation of protein abundance of immune checkpoint regulators by USP8 specific to PD-L1? Is the protein level of PD-1 regulated by USP8?

Reviewer #2 (Remarks to the Author):

USP8 inhibition reshapes an inflamed tumor microenvironment that potentiates the immunotherapy

In this manuscript, authors claim for a novel function of USP8 in control of PD-L1 abundance and immune modulation, with implications for innate immune response and MHC1 expression. Authors provide data to suggest that USP8 inhibition increases TRAF6-mediated K63-linked PD-L1 ubiquitination, which antagonizes K48-linked ubiquitination-dependent degradation. Authors also suggest that increases K63-linked TRAF6 ubiquitination following USP8 depletion or inhibition, results in TRAF6-mediated NF- κ B activation and elevates innate immune response/MHC1 expression. Based on these observations authors have tested the therapeutic efficacy of USP8 inhibition in combination with PD-1/PD-L1 blockade. Such combination treatment was found to attenuate tumor growth and extend the survival in syngeneic mouse models, which was not seen following either treatment alone.

Given the importance of identifying new targets for improved therapeutic response to immune checkpoint therapy, this study is timely and potentially important. However, experimental approaches used by authors to validate the underlying hypothesis appear to be somewhat biased, while authors conclusions appear preliminary, not fully supported by data presented.

1. Many experiments have been conducted in different type of cell lines and in syngeneic models. The hypothesis should be established and validated in the same model system. Because CT26, PC9, and H460 are extensively used throughout the study, the proposed mechanisms need to be validated in corresponding (unified) models.

a. CHX-chase experiment needed to be added to Fig. 1

b. Interaction and ubiquitination studies in Fig. 2B and 2H

c. Fig. 3D using CT26

d. Fig6. G-J using CT26 tumors

2. Authors claim an increase of PD-L1 abundance and NF- κ B activation by USP8 inhibition as mechanism that enhances the therapeutic efficacy of PD-1/PD-L1 blockade. However, there is no direct evidence in support of the functional importance of PD-L1 abundance and NF- κ B activation for the anti-tumor effect of USP8 inhibition. This point needs to be experimentally demonstrated; how the increase of PD-L1 abundance enhances the therapeutic effect of immune checkpoint blockades and the functional importance of USP8-control of NF- κ B activation in mouse models.

3. Given the increase in protein abundance, the transcript level of PD-L1 should be assessed in USP8 KO and TRAF6 KO models. Experimental data that excludes other mechanisms (e.g., autophagy, lysosome dependent degradation) for the regulation of PD-L1 abundance upon USP8 inhibition, should be provided.

4. Most experiments for PD-L1 modification by TRAF6 or USP8 were conducted using ectopic overexpression approaches in HEK293T cells. Authors need to demonstrate whether genetic depletion and chemical inhibition of USP8 or TRAF6 in CT26 cells affect K63-linked ubiquitination of PD-L1.

5. PD-L1 binds to USP8 through the cytoplasmic tail and TRAF6 via the TRAF6-binding motif in the extracellular region. Further discussion is required to address these molecular interactions upon altered conformation expected by post translational modification (i.e. ubiquitination discussed here) of these domains.

6. Authors claim that K63-linked ubiquitination of PD-L1 (TRAF6 gain, or USP8 inhibition) antagonizes K48-linked ubiquitination and increases PD-L1 abundance. The evidence to support the term "antagonizes" is required.

7. Does TRAF6 KO abolishes USP8 KO-driven effect on PD-L1 abundance? MHC1 expression? anti-tumor effect? in mouse models.

8. The results from RNAseq data should be validated experimentally. Particularly, surface expression of MHC1 and its functionality in USP8 KO should be validated. Likewise, immune cell infiltration, supporting the anti-tumor effect of USP8 inhibition, should be tested in syngeneic mouse models.

9. Given that USP8 inhibitor may affect immune cells and tumor cells, authors need to discuss possible consequences of systemic USP8 inhibition, compared to tumor-specific inhibition. Along these lines authors need to test and demonstrate how genetic depletion of USP8 or gain of TRAF6 enhances the therapeutic efficacy of PD-1/PD-L1 blockade.

10. In tumors (CT26 tumors and KP tumors) obtained in Fig. 6, the expression of PD-L1 and MHC1

should be assessed to corroborate authors hypothesis.

11. PD-L1 vs. USP8 IHC staining should be performed in lung squamous carcinoma (Fig. 3E) and CTL signature in TRAF6 low/high in colorectal cancer (Fig. 5F).

12. Data should be subjected to rigorous statistical analysis using the corresponding methods for each of the different studies performed. Data per number of experiments, reproducibility, statistical power analyses needed.

Point-to-point response to reviewers' comments (NCOMMS-21-29888-T)

We sincerely appreciate the thorough analyses and constructive suggestions provided by the two reviewers, which have been very helpful in guiding us to further improve our study. With extensive revision during the past more than four months, as described in more details below, we have obtained a substantial amount of new experimental evidence (21 new main figure panels and 64 new supplementary figure panels) to further strengthen our study, and to fully address all the concerns raised by the reviewers. We hope the editor and the reviewers will concur with us, after reading the enclosed point-to-point response, that we have experimentally addressed all the raised concerns in a satisfactory manner, and that the revised manuscript is now suitable for its publication as a research article in *Nature Communications*.

Reviewer #1 (Remarks to the Author):

Cancer immunotherapies targeting the PD-1/PD-L1 axis have demonstrated remarkable efficacy in a wide range of cancers. However, the molecular basis for resistance to PD-1/PD-L1 blockade is not fully understood. In this manuscript, Xiong and co-workers report a molecular mechanism by which USP8 regulates PD-L1 K63-linked ubiquitination and immune signalings to control anti-tumor immunity. They showed that inhibiting or depleting the deubiquitinase USP8 increases PD-L1 expression level through elevating TRAF6-mediated K63-linked ubiquitination of PD-L1, which may antagonize K48-linked ubiquitination and degradation. They also found that USP8 inhibition triggers innate immune responses through NF- κ B signaling. Taken together, their study indicates a potential combined therapeutic strategy that utilizes a USP8 inhibitor and PD1/PD-L1 blockade for enhancing efficacy of anti-tumor immunotherapies.

Overall, the authors provide clear evidence that USP8 regulates PD-L1 abundance and immune responses, placing USP8 as an important target for improving the efficacy of anti-PD-1/PD-L1 immunotherapies. The phenotypic data are compelling, including analysis of PD-L1 protein degradation, gene expression analysis using RNA-seq, and tumor growth in vivo. The relevance of USP8 inhibition in improving efficacy of anti-tumor immunotherapies will attract broad readers. My concerns relate to the proposed mechanism, e.g., USP8 is known to regulate the endosome-lysosomal pathway, but it is not clear from the data whether UPS8 regulates PD-L1 degradation through the proteasome or lysosome (see point 1 and 2). The authors should clarify the mechanism of PD-L1 degradation by adding more convincing data.

Response: We sincerely thank the reviewer for recognizing the novelty and the potential significant impact of this study. We also thank the reviewer for acknowledging our efforts in providing clear evidence and compelling phenotypic data to support our major conclusion that USP8 as an important target for improving the efficacy of anti-PD-1/PD-L1 immunotherapies.

At the same time, we thank the reviewer very much for raising several critical concerns and insightful suggestions, which have significantly improved our study. Following the reviewer's kind instructions, we have obtained more convincing results to experimentally address all the concerns raised by the reviewer as listed in our point-to-point responses below.

Major concerns:

1. USP8 is well known as a regulator of the endosomal sorting. Moreover, PD-L1 is a transmembrane protein, raising a possibility that USP8 may regulate degradation of PD-L1 through the endosome-lysosomal pathway. Although the involvement of USP8 in the regulating PD-L1 abundance is convincing (Fig. 1), it is not clear from the data presented whether the degradation is mediated by the proteasome or lysosome.

Response: We thank the reviewer for raising this concern. As kindly instructed, we have performed the following experiments to further clarify that the USP8-mediated regulation of PD-L1 stability is largely dependent on the proteasome system.

- a) Firstly, we generated the lung cancer PC9 stable cell lines that ectopically expressing the USP8 as well as GFP as a negative control using the lentivirus infection system. Consistent with our finding that USP8 negatively regulates the PD-L1 stability, we observed that ectopic expression of USP8, but not the GFP, dramatically decreased the endogenous PD-L1 protein abundance in PC9 cells (Supplementary Fig. 1M). Notably, although both the proteasome inhibitor MG132 and the lysosome inhibitor bafilomycin A1 (BafA1) treatment could elevate the PD-L1 protein abundance compared with control dimethyl sulfoxide (DMSO) treatment, MG132 had a more potent ability to rescue the USP8-mediated degradation of PD-L1 compared with the BafA1 (Supplementary Fig. 1O).
- b) Secondly, we used MG132, BafA1 as well as the DMSO as a control to treat the sgControl and sg*Usp8* CT26 cells. With BafA1 treatment, we still observed that there was an upregulation of PD-L1 protein level in sg*Usp8* (*Usp8* KO) cells compared with sgControl (WT) cells (Supplementary Fig. 1N). However, MG132 treatment almost alleviated the difference of PD-L1 expression between *Usp8* WT and KO cells (Supplementary Fig. 1N).
- c) Lastly, to further confirm that USP8-mediated regulation of PD-L1 stability was via the proteasome-dependent degradation, we used the protein synthesis inhibitor cycloheximide (CHX) to treat the sgControl and sg*Usp8* CT26 cells with/without additional MG132 treatment and analyzed the half-life changes of PD-L1. Without MG132 treatment, we observed that the half-life of PD-L1 protein is around 6 hours in sgControl cells, whereas the half-life of PD-L1 protein is significantly prolonged in sg*Usp8* cells (Fig. 1L and M). However, upon MG132 treatment, the difference of the PD-L1 protein half-life between the sgControl and sg*Usp8* cells was almost alleviated (Supplementary Fig. 1R and S).

Together, these results demonstrate that the USP8-governed degradation of PD-L1 is largely mediated by the proteasome system, at least in the experimental setting we tested.

2. Related to the point 1, the proteasomal inhibitor MG132 was added to the most of cell-based ubiquitination analyses (Fig. 2 and Fig. 3). However, MG132-dependency of PD-L1 ubiquitination is not clear from the data. Can the ubiquitination of PD-L1 be detected only under proteasomal inhibition? Do lysosomal inhibitors such as bafilomycin A affect PD-L1 ubiquitination?

Response: We thank the reviewer for raising this concern and also for providing insightful suggestions. As kindly suggested, we have tested how the proteasomal inhibitor MG132 or the lysosomal inhibitor bafilomycin A1 (BafA1) affects the status of PD-L1 ubiquitination and found that MG132 could dramatically elevate the ubiquitination of PD-L1 in cells compared with BafA1 treatment.

- a) We firstly co-transfected the pcDNA3.0-PD-L1 with His-tagged ubiquitin (His-Ub) into 293T cells as indicated and analyzed the ubiquitination of PD-L1 in denatured buffer after treatment with/without MG132 or BafA1, respectively. Our results showed that PD-L1 could be heavily modified with the ubiquitination upon MG132 treatment, whereas BafA1 treatment could slightly enhance the ubiquitination of PD-L1 in cells (Supplementary Fig. 1P).
- b) Furthermore, the treatment of MG132, but not BafA1, could partially rescue the USP8-dependent removal of PD-L1 ubiquitination in cells (Supplementary Fig. 1Q). Moreover, MG132 treatment, but not BafA1, dramatically elevated the TRAF6-mediated PD-L1 ubiquitination (Supplementary Fig. 3M).

These results together suggest that the proteasomal inhibition might be the major driver for affecting the status of PD-L1 ubiquitination in cells, at least under our experimental system.

3. The authors use ubiquitin mutants (K48 or K63 only) to analyze PD-L1-conjugated ubiquitin chains. However, such experiments can sometimes be artificial, and the results should be validated using other experimental settings such as the use of linkage-specific antibodies (or, ideally, mass spectrometric analyses). For example, the authors claimed differential regulation of K48- and K63-linked ubiquitin chains by USP8 (Fig. 2I). These experiments should also be performed using linkage-specific antibodies, such as Apu2 and Apu3, to detect endogenous Ub-derived chains.

Response: We thank the reviewer for the great suggestion. As the reviewer kindly suggested, we have performed the following parallel experiments to further validate the USP8-mediated removal of the K63-linked ubiquitin chain and promotion of the K48-linked ubiquitination on PD-L1 by immunoblotting using the K48- or K63-linkage-specific polyubiquitin antibody, respectively.

- a) Firstly, using the K48- or K63-linkage-specific polyubiquitin antibodies, we observed that gradually ectopic expression of USP8 resulted in decreasing the K63-linked ubiquitination of PD-L1 and increasing the PD-L1 K48-linked ubiquitination in 293T cells (Supplementary Fig. 2H and I). Moreover, stable ectopic expression of USP8 decreased the endogenous K63-linked ubiquitination of PD-L1 and increased the K48-linked ubiquitination of PD-L1 in CT26 and PC9 cells (Supplementary Fig. 2M and N).
- b) Secondly, inhibition of USP8 by genetic depletion or the pharmacological inhibitor treatment obviously upregulated the endogenous K63-linked ubiquitination, accompanying with reduced K48-linked ubiquitination of PD-L1 detecting by immunoblotting using the K48- or K63-linkage-specific polyubiquitin antibodies in the CT26 cells (Fig. 2L, Supplementary Fig. 2L).

- c) Thirdly, inhibition of TRAF6 by sgRNAs reduced the endogenous K63-linked ubiquitination of PD-L1 and increased the K48-linked ubiquitination of PD-L1 detecting by immunoblotting using the K48- or K63-linkage-specific polyubiquitin antibodies in CT26 cells (Supplementary Fig. 3O).
- d) Lastly, stable ectopic expression of TRAF6 promoted the endogenous K63-linked ubiquitination of PD-L1, accompanying with decreased K48-linked ubiquitination of PD-L1 detecting by immunoblotting using the K48- or K63-linkage-specific polyubiquitin antibodies in CT26 cells (Supplementary Fig. 3P).

Taken together, these results confirm that differential regulation of K48- and K63-linked ubiquitin chains on PD-L1 by USP8/TRAF6 can be observed at the endogenous levels using the K48- or K63-linkage-specific polyubiquitin antibodies, respectively.

4. It has been reported that PD-L1 gene is induced by NF- κ B at the transcription level. Therefore, the presented data that USP8 also regulates NF- κ B signaling is a bit confusing to me. This may imply that USP8 regulates PD-L1 protein abundance, at least in part, through activating NF- κ B and PD-L1 gene induction. This apparent discrepancy should be clarified.

Response: We thank the reviewer for raising this concern. In our current study, our results showed that USP8 regulates the NF- κ B signaling largely through interaction with TRAF6 and removal of K63-linked ubiquitination of TRAF6. As the reviewer speculated, USP8 might also regulate the PD-L1 gene transcription through the NF- κ B signaling pathway. However, our results showed although depletion of *Usp8* using the sgRNAs upregulated PD-L1 protein abundance and activated the NF- κ B signaling, the mRNA level of PD-L1 did not significantly change (Fig. 1H and K, Fig. 5J and K). Furthermore, we observed that the reported inhibitor, IKK16 (Waelchli R. et al., *Bioorg Med Chem Lett.*, 2006, 16(1):108-112; Moon C.S. et al., *Nat. Cancer*, 2021, 2(1):98-113) significantly suppressed the USP8-deficiency driven NF- κ B activation evidenced by decreasing the phosphorylation of p65 (Fig. 5J). However, IKK16 treatment did not inhibit the upregulation of PD-L1 protein abundance in *Usp8*-deficient CT26 cells (Fig. 5J). To further confirm this result, we have applied two independent shRNAs to deplete the endogenous *p65*, an important component for activation of NF- κ B signaling pathway, in *Usp8*-deficient CT26 cells. Our results showed that *p65* deficiency could significantly decrease the MHCI, but not the PD-L1, at both protein and mRNA levels in *Usp8*-depleted CT26 cells (Fig. 5L-O, Supplementary Fig. 5Q and R).

Moreover, depletion of *TRAF6* using the sgRNAs could markedly reduce the PD-L1 protein abundance, but did not significantly affect the PD-L1 mRNA level in multiple cancer cell lines (Fig. 3D-G, Supplementary Fig. 3E and F). These results together suggest that the NF- κ B signaling pathway might not be the major driver for PD-L1 gene transcription, at least in our experimental setting, which supports our conclusion that the USP8/TRAF6 regulates the PD-L1 protein abundance largely via the ubiquitination at the posttranslational level.

We agree with the reviewer that it has been previously reported that the PD-L1 gene transcription can be induced by the NF- κ B activation upon different stimulations at some specific

cancer settings. However, in addition to the induction of PD-L1 gene transcription by the NF- κ B signaling pathway, various upstream signaling pathways including, but not limited to, the IFN-JAK-STAT1, c-Myc, HIF1 α and Yap1/TAZ can regulate the PD-L1 expression at the transcriptional level, which was summarized in several review articles from our and other groups (Zhang J. et al., *Trends Biochem Sci.*, 2018, 43(12):1014-1032; Sun C. et al., *Immunity*, 2018, 48(3):434-452; Cha JH., *Mol. Cell*, 2019, 76(3):359-370). Thus, the regulation of PD-L1 gene transcription by various upstream transcriptional factors including NF- κ B should be context- or cancer type-dependent. While it might be outside the major scope of this current manuscript to identify how these transcriptional factors regulate the PD-L1 gene transcription in different cancer types, that awaits further investigation in the near future in a separate manuscript.

5. The authors show the direct interaction between USP8-PD-L1 (Fig. 2) and between TRAF6-PD-L1 (Fig. 3). They also show that USP8 and TRAF6 associate in cells (Fig. 5G). This raises a question whether USP8-PD-L1-TRAF6 form a ternary complex, or their interactions are antagonistic.

Response: We thank the reviewer for raising the insightful question. Our results have demonstrated that both USP8 and PD-L1 bind the coiled coil (CC) domain of TRAF6 (Fig. 3L and M, Fig. 5I), we hypothesized that their interactions might be antagonistic. To test this hypothesis, we have co-transfected the TRAF6 and PD-L1 constructs with gradually increasing amount of USP8 plasmids into 293T cells and detected how USP8 expression affects the interaction of TRAF6 with PD-L1. Our result demonstrated that increasingly ectopic expression of USP8 decreased the TRAF6 interaction with PD-L1 (Supplementary Fig. 5L). Moreover, gradually increasing expression of TRAF6 or PD-L1 also disrupted the interactions between the other two proteins interaction in cells (Supplementary Fig. 5M and N).

These data indicate that the interaction among USP8, PD-L1 and TRAF6 might be antagonistic in cells. Meanwhile, we do not exclude the USP8-PD-L1-TRAF6 form a transient ternary complex or other mechanisms that affect their interactions. As the reviewer #2 mentioned, the post-translational modification including ubiquitination on the PD-L1 or TRAF6 might alter their conformation to affect their interactions. We have discussed these possibilities on the page # 15 of revised manuscript.

Minor point:

6. Is the regulation of protein abundance of immune checkpoint regulators by USP8 specific to PD-L1? Is the protein level of PD-1 regulated by USP8?

Response: We thank the reviewer for raising these outstanding questions. As kindly instructed, we have examined how USP8 expression affects the protein abundance of other immune checkpoints. We firstly generated the PC9 stable cell lines that ectopically expressing USP8 as well as the empty vector (EV) as control using the lentivirus infection system. We found that ectopic expression of

USP8 dramatically decreased PD-L1 protein abundance, but did not affect the protein levels of other immune checkpoints including B7-H3, VISTA and CD47 (Supplementary Fig. 1M). Furthermore, USP8 inhibition using DUBs-IN-2 only elevated the protein abundance of PD-L1, but not that of B7-H3, VISTA, or CD47 immune checkpoint in PC9 cells (Supplementary Fig. 1E). Moreover, depletion of *USP8* using several independent shRNAs or inhibition of USP8 by the DUBs-IN-2 in MOLT4 or Jurkat cells did not affect PD-1 protein abundance (Supplementary Fig. 1F, G and J). These results together demonstrate that USP8 is likely a specific regulator for PD-L1, but not for other immune checkpoints we examined, including B7-H3, VISTA, CD47 or PD-1 in cells.

Reviewer #2 (Remarks to the Author):

In this manuscript, authors claim for a novel function of USP8 in control of PD-L1 abundance and immune modulation, with implications for innate immune response and MHCI expression. Authors provide data to suggest that USP8 inhibition increases TRAF6-mediated K63-linked PD-L1 ubiquitination, which antagonizes K48-linked ubiquitination-dependent degradation. Authors also suggest that increases K63-linked TRAF6 ubiquitination following USP8 depletion or inhibition, results in TRAF6-mediated NF- κ B activation and elevates innate immune response/MHCI expression. Based on these observations authors have tested the therapeutic efficacy of USP8 inhibition in combination with PD-1/PD-L1 blockade. Such combination treatment was found to attenuate tumor growth and extend the survival in syngeneic mouse models, which was not seen following either treatment alone.

Given the importance of identifying new targets for improved therapeutic response to immune checkpoint therapy, this study is timely and potentially important. However, experimental approaches used by authors to validate the underlying hypothesis appear to be somewhat biased, while authors conclusions appear preliminary, not fully supported by data presented.

Response: We sincerely thank the reviewer for acknowledging the novelty and the potential significant impact of our study. We also thank the reviewer very much for pointing out concerns and providing us insightful suggestions, which have helped us to further improve our study. As the reviewer kindly instructed and suggested, we have thoroughly revised our manuscript and experimentally addressed all the concerns raised by the reviewer as listed in our point-to-point responses below.

1. Many experiments have been conducted in different type of cell lines and in syngeneic models. The hypothesis should be established and validated in the same model system. Because CT26, PC9, and H460 are extensively used throughout the study, the proposed mechanisms need to be validated in corresponding (unified) models.
 - a. CHX-chase experiment needed to be added to Fig. 1
 - b. Interaction and ubiquitination studies in Fig. 2B and 2H
 - c. Fig. 3D using CT26
 - d. Fig. 6H-J using CT26 tumors

Response: We thank the reviewer for raising the concern and providing us excellent suggestions. We totally agree with the reviewer that we performed most of experiments in the setting of lung cancer cells (PC9 and H460) and colon cancer cells (CT26) throughout the study. Thus, we have performed following experiments to further validate our findings in these cells as the reviewer suggested.

- a. We performed the cycloheximide (CHX)-chase experiment in sgControl and knockout (sg*Usp8*) CT26 cells and also found that the half-life of PD-L1 protein was significantly extended in *Usp8*-deficient cells compared with the WT cells (Fig. 1L and M).
 - b. The immunoprecipitation (IP) assays showed that the interaction between USP8 and PD-L1 could also be detected at the endogenous levels in CT26 and PC9 cells (Fig. 2B and 2C). Moreover, the *in vivo* ubiquitination assays demonstrated that the K63-linked ubiquitination of PD-L1 could also be observed in both CT26 and PC9 cells (Fig. 2I and J).
 - c. As kindly suggested, we depleted *Traf6* in CT26 cells using sgRNAs and also found that TRAF6 deficiency dramatically decreased PD-L1 protein abundance, but did not significantly affect the PD-L1 mRNA level (Fig. 3D-G).
 - d. We have analyzed various tumor-infiltrating lymphocytes (TILs) in CT26 tumors from immunocompetent BALB/c mice treated with the USP8 inhibitor, anti-PD-L1 antibody, alone or combination (Fig. 6H-J and Supplementary Fig. 6F-H).
2. Authors claim an increase of PD-L1 abundance and NF- κ B activation by USP8 inhibition as mechanism that enhances the therapeutic efficacy of PD-1/PD-L1 blockade. However, there is no direct evidence in support of the functional importance of PD-L1 abundance and NF- κ B activation for the anti-tumor effect of USP8 inhibition. This point needs to be experimentally demonstrated; how the increase of PD-L1 abundance enhances the therapeutic effect of immune checkpoint blockades and the functional importance of USP8-control of NF- κ B activation in mouse models.

Response: We thank the reviewer for the insightful question and also for providing the constructive suggestion. To address this question, we have performed the following key experiments.

- a. To examine the role of USP8 inhibition mediated upregulation of PD-L1 protein abundance in regulating the therapeutic efficacy of immune checkpoint blockades, we treated the immunocompetent BALB/c mice bearing *Pd-11* KO CT26 (Dai X. et al., *Mol. Cell*, 2021, 81(11):2317-2331) tumors with the anti-PD-L1 antibody, USP8 inhibitor, alone or combination. Our results demonstrated that there was no significant difference in tumor growth between USP8 inhibitor alone and its combination with anti-PD-L1 antibody treatment in *Pd-11* KO CT26 tumor-bearing mice (Supplementary Fig. 7M). However, we observed that USP8 inhibitor treatment significantly elevated the phosphorylation-p65 (p-p65) and cellular surface of MHC-I expression, indicating that USP8 inhibition could still activate the NF- κ B signaling in *Pd-11* KO CT26 tumor cells (Supplementary Fig. 7N and O). Moreover, the USP8 inhibitor treatment also significantly enhanced the infiltrating cytotoxic CD8⁺ T cells in *Pd-11* KO CT26 tumor microenvironment (Supplementary Fig. 7P-R). These results support the notion that the upregulation of PD-L1 protein abundance by USP8 inhibition is also required for enhancing the therapeutic effect of anti-PD-L1 immunotherapy.
- b. To further address the functional importance of NF- κ B activation by the USP8 inhibition in regulating the therapeutic efficacy of PD-L1 blockade, we further applied the shRNAs to

deplete the endogenous *p65*, an important component for NF- κ B signaling activation, in *Usp8*-deficient CT26 cells. We observed that *p65* deficiency significantly decreased MHC-I expression, but not that of PD-L1, at both protein and mRNA levels in *Usp8*-deficient CT26 cells (Fig. 5L-O). Next, we tested how the *Usp8*-deficient and *Usp8/p65* double deficient CT26 tumors responded to anti-PD-L1 immunotherapy *in vivo*. Our results showed that *Usp8* KO CT26 tumors sensitized to the PD-L1 blockade compared with *Usp8/p65* double deficient CT26 tumors upon the anti-PD-L1 immunotherapy (Supplementary Fig. 7S-U), which might be due to *p65* deficiency abolishing the *Usp8* KO-driven upregulation of MHC-I and tumor-infiltrating CD8⁺ cytotoxic T cells (Supplementary Fig. 7V-Y). These results indicate that the NF- κ B activation by USP8 inhibition is also needed to sensitize tumors to the anti-PD-L1 immunotherapy.

Taken together, these results suggest that an increase of PD-L1 protein abundance and NF- κ B activation by USP8 inhibition might reshape an inflamed tumor microenvironment, which enhances the therapeutic efficacy of PD-1/PD-L1 blockade *in vivo*.

3. Given the increase in protein abundance, the transcript level of PD-L1 should be assessed in USP8 KO and TRAF6 KO models. Experimental data that excludes other mechanisms (e.g., autophagy, lysosome dependent degradation) for the regulation of PD-L1 abundance upon USP8 inhibition, should be provided.

Response: We thank the reviewer for providing us these great suggestions. As kindly suggested, we have assessed the mRNA level of PD-L1 using quantitative RT-qPCR and found that depletion of *Usp8* using the sgRNAs dramatically upregulated PD-L1 protein abundance, but did not significantly affect the PD-L1 mRNA level in CT26 cells (Fig. 1H and K). Moreover, although PD-L1 protein abundance was dramatically decreased in TRAF6 KO H460 and CT26 cells compared with their corresponding control cells, the mRNA level was not significantly altered in TRAF6 KO compared with control cells (Fig. 3D-G). These results further support our conclusion that the USP8/TRAF6 regulates the PD-L1 stability might be largely dependent on the post-translational ubiquitination on PD-L1 protein.

As the reviewer mentioned, there are two main mechanisms for controlling protein stability and degradation in cells: the ubiquitin-proteasome system (UPS) and the autophagy-lysosome pathway. To exclude the autophagy-lysosome pathway involving in the regulation of PD-L1 protein abundance upon USP8 inhibition, we have applied the proteasome inhibitor MG132 and the autophagy/lysosome inhibitor bafilomycin A1 (BafA1) to treat the *Usp8* wide type (sgControl) and knockout (sg*Usp8*) CT26 cells. With BafA1 treatment, we still observed that there was an upregulation of PD-L1 protein level in *Usp8* KO cells compared with the WT CT26 cells (Supplementary Fig. 1N). However, the MG132 treatment almost alleviated the difference of PD-L1 expression between *Usp8* WT and KO CT26 cells (Supplementary Fig. 1N).

Moreover, we also generated the lung cancer PC9 stable cell lines that ectopically expressing the USP8 as well as GFP as a negative control using the lentiviral infection system. MG132 had a more potent ability to rescue the USP8-mediated degradation of PD-L1 compared

with BafA1 treatment (Supplementary Fig. 1O). These results suggest that USP8 regulates the degradation of PD-L1 largely through the proteasome system in cells.

4. Most experiments for PD-L1 modification by TRAF6 or USP8 were conducted using ectopic overexpression approaches in HEK293T cells. Authors need to demonstrate whether genetic depletion and chemical inhibition of USP8 or TRAF6 in CT26 cells affect K63-linked ubiquitination of PD-L1.

Response: We thank the reviewer for the concern and great suggestions. As kindly instructed, we firstly detected how USP8 deficiency affects the endogenous K63-linked ubiquitination status in CT26 cells. Inhibition of *Usp8* by sgRNA-mediated depletion or the pharmacological inhibitor DUBs-IN-2 obviously elevated the endogenous K63-linked ubiquitination of PD-L1, accompanying with reduced K48-linked ubiquitination of PD-L1 detecting by the K48- or K63-linkage-specific polyubiquitin antibodies in the CT26 cells (Fig. 2L, Supplementary Fig. 2L). On the other hand, stable ectopic expression of USP8 in CT26 or PC9 cells dramatically decreased the endogenous K63-linked ubiquitination of PD-L1 and increased the K48-linked ubiquitination of PD-L1 detecting by the K48- or K63-linkage-specific polyubiquitin antibodies (Supplementary Fig. 2M and N).

Furthermore, genetic depletion of *Traf6* using sgRNAs reduced the endogenous K63-linked ubiquitination of PD-L1 and increased the K48-linked ubiquitination of PD-L1 detecting by the K48- or K63-linkage-specific polyubiquitin antibodies in CT26 cells (Supplementary Fig. 3O). Conversely, stable ectopic expression of TRAF6 increased the endogenous K63-linked ubiquitination of PD-L1, accompanying with reduced K48-linked ubiquitination of PD-L1 detecting by the K48- or K63-linkage-specific polyubiquitin antibodies in CT26 cells (Supplementary Fig. 3P).

Taken together, these results further confirm that genetic or chemical manipulation of USP8/TRAF6 expression or activity could affect the PD-L1 K63-linked ubiquitination status at the endogenous levels.

5. PD-L1 binds to USP8 through the cytoplasmic tail and TRAF6 via the TRAF6-binding motif in the extracellular region. Further discussion is required to address these molecular interactions upon altered conformation expected by post translational modification (i.e. ubiquitination discussed here) of these domains.

Response: We thank the reviewer for the great suggestion. As kindly suggested, we have added the discussion that the PD-L1 ubiquitination status might alter its conformation to affect its interaction with USP8 or TRAF6 in cells in the revised manuscript. Our results also demonstrated that both USP8 and PD-L1 bind the coiled coil (CC) domain of TRAF6 (Fig. 3L and M, Fig. 5I), we hypothesized that their interactions might be competing or antagonistic. To test this, we co-transfected the TRAF6 and PD-L1 constructs with gradually increasing amount of USP8 plasmids into 293T cells and detected how USP8 expression affects the interaction of TRAF6 with PD-L1.

Our result demonstrated that increasingly ectopic expression of USP8 decreased the TRAF6 interaction with PD-L1 in cells (Supplementary Fig. 5L). Moreover, gradually increasing expression of TRAF6 or PD-L1 also could disrupt the other two protein interaction in cells (Supplementary Fig. 5M and N).

These data together indicate that the interaction among USP8, PD-L1 and TRAF6 might be competing or antagonistic in cells. Meanwhile, we do not exclude other mechanisms that affect their interactions. As the reviewer suggested, the post-translational modification including ubiquitination on the PD-L1 or TRAF6 might alter their conformation to affect their interactions. We have discussed these possibilities on the page # 15 of revised manuscript.

6. Authors claim that K63-linked ubiquitination of PD-L1 (TRAF6 gain, or USP8 inhibition) antagonizes K48-linked ubiquitination and increases PD-L1 abundance. The evidence to support the term “antagonizes” is required.

Response: We thank the reviewer for pointing out this concern. To address how K63-linked ubiquitination of PD-L1 (TRAF6 gain, or USP8 inhibition) antagonizes K48-linked ubiquitination, we have performed the following several parallel experiments.

- a) Firstly, under the ectopic expression system, we observed that ectopic expression of USP8 results in decreasing the K63-linked ubiquitination of PD-L1 and increasing the PD-L1 K48-linked ubiquitination detected by immunoblotting using the K48- or K63-linkage-specific polyubiquitin antibodies in 293T cells (Supplementary Fig. 2H and I). Moreover, stable ectopic expression of USP8 dramatically reduced the endogenous K63-linked ubiquitination of PD-L1, accompanying with increased K48-linked ubiquitination of PD-L1 detecting by the K48- or K63-linkage-specific polyubiquitin antibodies in CT26 or PC9 cells (Supplementary Fig. 2M and N).
- b) Secondly, inhibition of USP8 by the genetic depletion or the pharmacological inhibitor treatment obviously upregulated the endogenous K63-linked ubiquitination of PD-L1 in the CT26 cells, accompanying with reduced K48-linked ubiquitination of PD-L1 detected by immunoblotting using by the K48- or K63-linkage-specific polyubiquitin antibodies in CT26 cells (Fig. 2L, Supplementary Fig. 2L).
- c) Thirdly, inhibition of *Traf6* by the genetic depletion reduced the endogenous K63-linked ubiquitination of PD-L1 and increased the K48-linked ubiquitination of PD-L1 in CT26 cells (Supplementary Fig. 3O). Conversely, ectopic expression of TRAF6 promoted the endogenous K63-linked ubiquitination of PD-L1, accompanying with decreased K48-linked ubiquitination of PD-L1 in CT26 cells (Supplementary Fig. 3P).

Taken together, these results support that the endogenous K63-linked ubiquitination of PD-L1 antagonizes K48-linked ubiquitination, which might be dynamically regulated by USP8 and TRAF6 in cells.

7. Does TRAF6 KO abolishes USP8 KO-driven effect on PD-L1 abundance? MHC-I expression? anti-tumor effect? in mouse models.

Response: We thank the reviewer for raising these insightful comments. To address these questions, we have firstly generated the *Usp8/Traf6* double KO cells using the CRISPR-Cas9 technology to deplete *Traf6* in *Usp8* KO CT26 cells (Supplementary Fig. 7A). Our results showed that depletion of *Traf6* abolished *Usp8* KO-driven upregulation of PD-L1 protein abundance and the MHC-I expression in CT26 cells (Supplementary Fig. 7B-E). To further explore how the WT, *Usp8* KO and *Usp8/Traf6* double KO CT26 tumors responded to anti-PD-L1 immunotherapy in mouse model, we transplanted these genetically edited CT26 cells into the immunocompetent BALB/c mice and treated with the anti-PD-L1 antibodies. Our results demonstrated that *Traf6* deficiency almost abolished the *Usp8*-deficient CT26 tumors sensitization to the PD-L1 blockade in mouse tumor model (Supplementary Fig. 7F-H). Moreover, analyses of tumor microenvironment (TME) showed that expression of PD-L1 and MHC-I on surface of tumor cells, and the tumor-infiltrating CD8⁺ cytotoxic T cells were significantly decreased in *Usp8/Traf6* double KO CT26 tumors compared with *Usp8* KO CT26 tumors (Supplementary Fig. 7I-L). These results suggest that *Traf6* deficiency largely abolishes *Usp8* KO-driven anti-tumor effect via altering the TME.

8. The results from RNAseq data should be validated experimentally. Particularly, surface expression of MHC-I and its functionality in USP8 KO should be validated. Likewise, immune cell infiltration, supporting the anti-tumor effect of USP8 inhibition, should be tested in syngeneic mouse models.

Response: We thank the reviewer for raising these great suggestions. As kindly suggested, we have firstly confirmed that the cellular surface expression of MHC-I was also significantly elevated in *Usp8* KO CT26 cells compared with WT CT26 cells using the flow cytometry analysis (Fig. 5D and E). In keeping with our conclusion that USP8 regulates the MHC-I through the NK- κ B signaling pathway, we applied two independent shRNAs to deplete the endogenous *p65*, an important component for NF- κ B activation in *Usp8* KO CT26 cells and found that the *p65* deficiency significantly decreased the *Usp8* KO-driven upregulation of MHC-I, but not the expression of PD-L1 protein, on the surface of *Usp8* KO CT26 cells (Fig. 5L-N, Supplementary Fig. 5Q and R). These results demonstrate that depletion of *p65* could block the TRAF6-NF- κ B-MHC-I pathway. Thus, we examined whether depletion of *p65* could compromise the *Usp8*-deficient CT26 tumors response to anti-PD-L1 immunotherapy *in vivo*. Our results demonstrated that *Usp8*-deficient CT26 tumors sensitized to the PD-L1 blockade compared with *Usp8/p65* double deficient CT26 tumors upon the anti-PD-L1 immunotherapy (Supplementary Fig. 7S-U), which might be due to *p65* deficiency significantly reducing MHC-I expression and the number of tumor-infiltrating CD8⁺ cytotoxic T cells (Supplementary Fig. 7V-Y). These results indicate that USP8 inhibition activates the NF- κ B signal to upregulate surface expression of MHC-I is also required for sensitizing tumors to the anti-PD-L1 immunotherapy *in vivo*.

9. Given that USP8 inhibitor may affect immune cells and tumor cells, authors need to discuss possible consequences of systemic USP8 inhibition, compared to tumor-specific inhibition. Along these lines authors need to test and demonstrate how genetic depletion of USP8 or gain of TRAF6 enhances the therapeutic efficacy of PD-1/PD-L1 blockade.

Response: We thank the reviewer for raising this concern and great suggestion. To exclude the possible consequences of systemic USP8 inhibition, we have evaluated how genetic depletion of *Usp8* in CT26 tumor cells affected the therapeutic efficacy of PD-L1 blockade *in vivo*. To this end, we transplanted the *Usp8* WT and KO CT26 cells into the immunocompetent BALB/c mice and treated with/without anti-PD-L1 antibody. Our results demonstrated that the tumor growth from *Usp8* KO with anti-PD-L1 treatment group was significantly retarded compared with the group of *Usp8* WT with PD-L1 blocking or *Usp8* KO with control IgG treatment (Fig. 6K, Supplementary Fig. 6P and Q). Moreover, tumors from *Usp8* KO with anti-PD-L1 treatment group exhibited high levels of PD-L1, MHC-I, and tumor-infiltrating cytotoxic CD8⁺ T cells (Fig. 6L and M, Supplementary Fig. 6R and S). Although these results demonstrate that tumor-specific genetic inhibition of USP8 could significantly enhance the therapeutic efficacy of PD-L1 blockade, we also incorporate the discussion that the systemically using USP8 inhibitor may affect the function of other cells including immune cells on the page # 18 of revised manuscript as the reviewer kindly suggested.

10. In tumors (CT26 tumors and KP tumors) obtained in Fig. 6, the expression of PD-L1 and MHC I should be assessed to corroborate authors hypothesis.

Response: We thank the reviewer for this kind suggestion. As kindly suggested, we have performed the flow cytometry assay to detect the expression of PD-L1 and MHC-I in CT26 tumors treated with anti-PD-L1 antibody, USP8 inhibitor, alone or combination. Our results demonstrated that USP8 inhibitor alone or combinational treatment significantly elevated the cellular surface PD-L1 and MHC-I on tumor cells derived from CT26 tumors compared with the control treatment group (Supplementary Fig. 6I-K). In keeping with this finding, the USP8 inhibitor alone or combinational treatment also significantly upregulated the expression of PD-L1 and MHC-I on tumor samples derived from *Kras*^{G12D/+}*Tp53*^{fl/fl} (KP) mice compared with the control treatment group (Supplementary Fig. 6L-O).

11. PD-L1 vs. USP8 IHC staining should be performed in lung squamous carcinoma (Fig. 3E) and CTL signature in TRAF6 low/high in colorectal cancer (Fig. 5F).

Response: We thank the reviewer for this kind suggestion. As kindly suggested, we have performed the IHC staining for USP8 and PD-L1 in the patient tissues of lung squamous carcinoma (Original Fig. 3E, revised Fig. 3H). Our results showed that there was a negative co-relationship between the USP8 and PD-L1 in these samples of lung squamous carcinoma (Fig. 1Q, Supplementary Fig. 1T). We also analyzed the CTL signature in TRAF6 low/high in colorectal

cancer (Original Fig. 5F, revised Fig. 5H) and found there was a trend that cancer patients with high TRAF6 expression and a higher level of CTL had a better survival (Supplementary Fig. 5J).

12. Data should be subjected to rigorous statistical analysis using the corresponding methods for each of the different studies performed. Data per number of experiments, reproducibility, statistical power analyses needed.

Response: We thank the reviewer for raising these concerns. As you kindly instructed, we have added the detailed information of statistical analysis including the method used, the number of biological repeats, and statistical difference in the figure legends and the method part in the revised manuscript.

REVIEWERS' COMMENTS

Reviewer #1 (Remarks to the Author):

In the revised manuscript, the authors have addressed all the concerns raised by this reviewer. No further questions.

Reviewer #2 (Remarks to the Author):

Authors have adequately addressed reviewers comments, and as a result, the revised manuscript is substantially improved.

Point-to-point response to reviewers' comments (NCOMMS-21-29888-A)

Reviewer #1 (Remarks to the Author):

In the revised manuscript, the authors have addressed all the concerns raised by this reviewer. No further questions.

Response: We sincerely thank the reviewer for acknowledging our efforts to fully address all the concerns, which have substantially improved our manuscript.

Reviewer #2 (Remarks to the Author):

Authors have adequately addressed reviewers comments, and as a result, the revised manuscript is substantially improved.

Response: We thank the reviewer very much for recognizing our efforts to adequately address reviewers' comments, which have significantly improved our study.